# CAN MEDICAL VISION-LANGUAGE PRE-TRAINING SUCCEED WITH PURELY SYNTHETIC DATA?

## ABSTRACT

Medical Vision-Language Pre-training (MedVLP) has made significant progress in enabling zero-shot tasks for medical image understanding. However, training MedVLP models typically requires large-scale datasets with paired, high-quality image-text data, which are scarce in the medical domain. Recent advancements in Large Language Models (LLMs) and diffusion models have made it possible to generate large-scale synthetic image-text pairs. This raises the question: ***Can MedVLP succeed using purely synthetic data?*** To address this, we use off-the-shelf generative models to create synthetic radiology reports and paired Chest X-ray (CXR) images, and propose an automated pipeline to build a diverse, high-quality synthetic dataset, enabling a rigorous study that isolates model and training settings, focusing entirely from the data perspective. Our results show that MedVLP models trained *exclusively on synthetic data* outperform those trained on real data by **3.8%** in averaged AUC on zero-shot classification. Moreover, using a combination of synthetic and real data leads to a further improvement of **9.07%**. Additionally, MedVLP models trained on synthetic or mixed data consistently outperform those trained on real data in zero-shot grounding, as well as in fine-tuned classification and segmentation tasks. Our analysis suggests MedVLP trained on well-designed synthetic data can outperform models trained on real datasets, which may be limited by low-quality samples and long-tailed distributions[1].

## 1 INTRODUCTION

In medical image analysis, learning representative features typically requires labor-intensive and costly image annotations (Ronneberger et al., 2015; Liu et al., 2023b). Medical Vision-Language Pre-training (MedVLP) addresses this challenge by aligning vision and language content using paired datasets of images and clinical reports, reducing the need for manual annotations (Radford et al., 2021; Zhang et al., 2020; Wu et al., 2023; Liu et al., 2023a). However, existing MedVLP models rely heavily on large-scale, high-quality paired data (Liu et al., 2023e), which is scarce in practice. Real-world datasets often contain noisy data, such as low-quality images and unpaired image-text samples, degrading model performance (Xie et al., 2024; Bannur et al., 2023). Recent advancements in Large Language Models (LLMs) and diffusion models enable the generation of large-scale synthetic image-text datasets, offering an alternative to traditional data collection. Although these techniques have shown promise in medical tasks, they are primarily used as auxiliary support for real data via augmentation (Chen et al., 2024a; Yao et al., 2021; Chen et al., 2022; Qin et al., 2023), and are often limited to single-modality settings. To the best of our knowledge, no studies have fully explored the potential of using synthetic multimodal data for MedVLP or considered training exclusively on synthetic data (Liu et al., 2023e).

To bridge this gap and showcase synthetic data's potential for MedVLP, our contributions are:

- We propose an automated pipeline to create the **SynCXR** dataset, which contains 200,000 synthetic images and text generated with quality and distribution control using off-the-shelf models, without relying on real data or manual curation.

- We successfully demonstrate that MedVLP models trained on our SynCXR dataset, containing only synthetic data, outperform those trained on real data. Moreover, combining

---

[1]All data and code will be released upon acceptance.

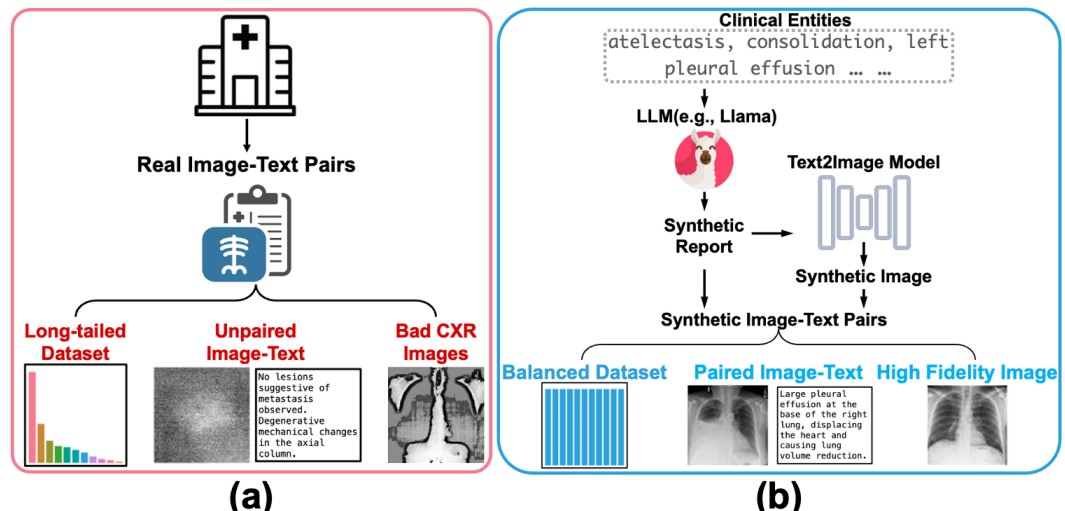

Figure 1: Comparison of real image-text datasets and synthetic datasets. **(a):** The real image-text dataset, MIMIC-CXR (Johnson et al., 2019b), while authentic, often contains imperfections such as long-tailed data distribution, unpaired images and text, and low-quality CXR images, which limit the performance of MedVLP models pretrained on this dataset. **(b):** The synthetic dataset generation process uses clinical entities as prompts to an LLM (e.g., Llama3.1 (AI@Meta, 2024)) to generate synthetic reports. These reports are then used to create synthetic images through RoentGen (Bluethgen et al., 2024). We propose an automated pipeline to control the dataset distribution, ensuring it is balanced and includes paired image-text samples.

    synthetic and real data further improves performance, showcasing the effectiveness of our synthetic data generation pipeline.

- We identify several issues in the most commonly used real dataset for MedVLP, MIMIC-CXR (Johnson et al., 2019b), that degrade MedVLP performance, including low-quality images and unpaired image-text samples. Furthermore, we identify the long-tailed distribution problem in multimodal datasets, as shown in Fig 1, 2.

- We conduct an extensive analysis of the key factors contributing to MedVLP's success using purely synthetic data. Our method is evaluated on seven downstream tasks using zero-shot learning and linear probing, demonstrating that MedVLP can effectively perform with synthetic data alone.

## 2 RELATED WORK

**Representation Learning with Synthetic Data.** Synthetic data has been widely employed across various deep learning fields (Rossenbach et al., 2020; Varol et al., 2017; Jahanian et al., 2022; Zhou et al., 2023; Yang et al., 2020; Li et al., 2023). In visual representation learning, synthetic data has improved model performance in a range of tasks (Richter et al., 2016; Ros et al., 2016; Chen et al., 2019; Johnson-Roberson et al., 2017; Yuan et al., 2024; Shmelkov et al., 2018). Recent efforts have also focused on using synthetic data from text-to-image models to augment real-world data during training (Azizi et al., 2023; Sariyildiz et al., 2023; He et al., 2023). For example, (Yu et al., 2023) introduced a framework to generate synthetic images to diversify existing datasets. Notably, methods utilizing text-to-image generative models (Rombach et al., 2022) have demonstrated that synthetic images guided by real captions can effectively train self-supervised models, achieving performance comparable to that of real images (Tian et al., 2023b).

Further advancements like SynCLR (Tian et al., 2023a) have focused on visual representation learning using only synthetic images, generated with conditioning on various categories. Meanwhile, other recent works (Fan et al., 2023; Sharifzadeh et al., 2024; Xie et al., 2024) have explored joint image and text generation for enhanced vision-language pretraining (VLP). However, only one study, SynthCLIP (Hammoud et al., 2024), investigates VLP exclusively with synthetic data, and even that work is limited to natural images. To date, no research has explored the potential of MedVLP trained solely on synthetic data.

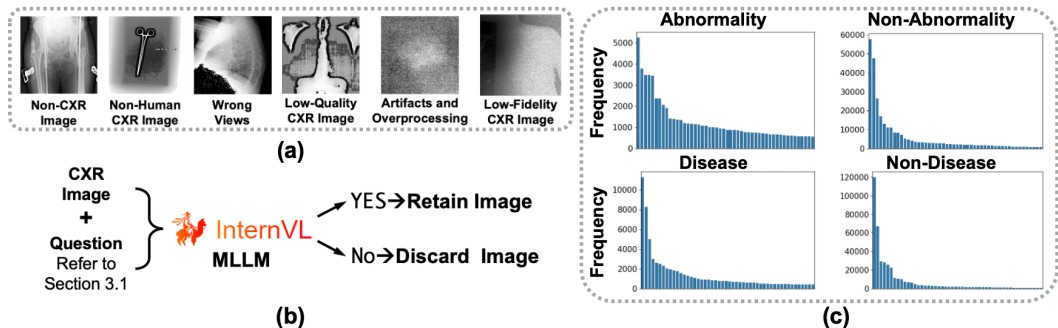

Figure 2: **(a):** Examples of invalid or low-quality images filtered out by the proposed image curation method described in Sec 3.1. **(b):** The image curation pipeline uses InternVL2 (Chen et al., 2023), a Multimodal Large Language Model (MLLM), to assess CXR image quality. Images that meet the criteria are retained; others are discarded. **(c):** Entity frequency distribution in the MIMIC-CXR dataset. Due to space constraints, only the top 50 frequent entities for four categories (Abnormality, Non-Abnormality, Disease, Non-Disease) are shown. A more detailed distribution is presented in Fig 6,7,10,8,9.

**Medical Vision Language Pre-training.** Recent work on MedVLP has focused on integrating visual and textual modalities, particularly for chest X-ray (CXR) images. Studies such as (Zhang et al., 2020; Huang et al., 2021; Wang et al., 2022; Liu et al., 2023b;d;c; Wan et al., 2024) have concentrated on aligning CXR images with paired radiology reports. Some methods also leverage external datasets to boost performance, raising concerns about generalizability (Wu et al., 2023; Zhang et al., 2023; Li et al., 2024; Phan et al., 2024a). However, all current MedVLP approaches rely heavily on large-scale, real image-text paired datasets like MIMIC-CXR (Johnson et al., 2019b). Some even require additional human-annotated datasets or manual interventions to improve model performance (Wu et al., 2023; Zhang et al., 2023; Phan et al., 2024a), which limits their scalability and accessibility.

**Synthetic Data for Medical Image Tasks.** Given the scarcity of annotated data, high costs, and privacy concerns in medical data collection, synthetic data has been explored to support various medical image tasks (Koetzier et al., 2024). However, most prior work focuses on image modality and supervised learning (Chen et al., 2024a; Yao et al., 2021; Chen et al., 2022; Qin et al., 2023), using synthetic data solely as augmentation for real datasets (Khosravi et al., 2024; Ktena et al., 2024). Few studies have evaluated models trained entirely on synthetic medical data (Wu et al., 2024). Recent efforts have generated synthetic text and images for MedVLP (Xie et al., 2024), but still restrict synthetic data usage to augmentation. Consequently, the full potential of synthetic data in MedVLP remains largely unexplored.

In this work, we generate both synthetic CXR images and reports, then training a MedVLP model solely on synthetic data. We conduct an extensive evaluation of the impact of large-scale synthetic medical data on MedVLP, exploring its performance across various downstream tasks.

## 3 METHODS

### 3.1 EXPLORING IMPERFECTIONS IN REAL DATA

For MedVLP, the most commonly used dataset is MIMIC-CXR (Johnson et al., 2019a;b), a collection of chest x-ray (CXR) images paired with their corresponding textual reports. after following the preprocessing steps outlined in previous works (Zhang et al., 2023; Wang et al., 2022; Huang et al., 2021), this dataset provides a total of 213,384 image-text pairs for pre-training. And all images must be frontal views according to the preprocessing steps outlined in (Huang et al., 2021).

Previous work on VLP with natural images (Xu et al., 2023b) has shown that data quality, including image fidelity and long-tailed distribution, significantly impacts model performance. However, the quality of MedVLP datasets remains underexplored due to ambiguity in defining medical image quality, stemming from diverse imaging protocols. Additionally, quantifying data distribution is complex, as radiology reports often describe patterns across multiple anatomical regions rather than distinct categories. To address these challenges, we develop a systematic pipeline to thoroughly

analyze the data issues in the MIMIC-CXR (Johnson et al., 2019b) dataset, as detailed in the following sections.

**Low-Quality and Mismatched Image-Text Pairs.** Our aim is to explore and identify issues related to image quality in the MIMIC-CXR dataset (Johnson et al., 2019a), rather than to completely clean the dataset, as creating a perfect dataset and filtering out all low-quality samples is infeasible for large-scale multimodal datasets (Xu et al., 2023a). Inspired by (Bannur et al., 2023), which highlights various issues with poor-quality images, we design six queries for a Multimodal Large Language Model (MLLM), utilizing the InternVL2-26B model[2] (Chen et al., 2023; 2024b). Each CXR image from the MIMIC-CXR dataset is paired with these six queries, and the MLLM process each query independently. The process is depicted in Fig 2 (b).

- **Detecting Non-CXR Images:** `<CXR Image>, Please check if the given image is a chest X-ray scan. If it is a chest X-ray, return 'YES'. Otherwise, return 'NO'.`

- **Detecting Non-Human CXR Images:** `<CXR Image>, Please verify if the given image is a human chest X-ray scan. If it is a chest X-ray, return 'YES'. Otherwise, return 'NO'.`

- **Detecting Wrong Views:** `<CXR Image>, Please check if the given image is a frontal chest X-ray view. If it is a frontal view, return 'YES'. If it is a lateral view or any other view, return 'NO'.`

- **Assessing Image Quality:** `<CXR Image>, Please analyze the provided chest X-ray (CXR) image and respond with 'NO' if the image quality is poor, such as being blurry, containing artifacts, or having poor contrast. Respond with 'YES' if the image quality is acceptable.`

- **Detecting Artifacts and Overprocessing:** `<CXR Image>, Please analyze the following chest X-ray image. Respond with 'YES' if the image is clear, correctly oriented, and free of artifacts or imperfections that could affect its diagnostic quality. Respond with 'NO' if the image is blurry, incorrectly oriented, contains artifacts, or has imperfections that make it unsuitable for further analysis.`

- **Checking High-Fidelity:** `<CXR Image>, Please check if the given image is a high-fidelity human chest X-ray scan. If it is a high-fidelity chest X-ray, return 'YES'. Otherwise, return 'NO'.`

After this process, we filter out the CXR images where the answers are all `'NO'` across the six queries. Fig 2 (a) shows examples of images where the answer was `'NO'`. We identified and removed 1,448 such images and their corresponding reports from the preprocessed MIMIC-CXR dataset, leaving us with 211,936 image-text pairs.

To further refine the dataset, we use the CXR-specific vision encoder, RAD-DINO (Pérez-García et al., 2024), to extract image features from the remaining 211,936 CXR images and from the 1,448 samples identified as bad by MLLM filtering. We then compute the similarity between each image in the cleaned dataset and each of the bad samples. Since each image comes from a different clinical case, we only compare image quality rather than the clinical content (e.g., diagnoses or abnormalities). To do this, we set a similarity threshold of 0.5 and remove all images with a similarity score greater than 0.5. This step resulted in the removal of an additional 5,512 images and their paired reports, reducing the dataset to 206,424 image-text pairs. Fig 2 (a) also shows the samples removed based on their similarity to bad images using visual features from RAD-DINO[3] (Pérez-García et al., 2024).

In our exploration of the MIMIC-CXR dataset, we utilized a rough approach to identify problematic images, such as non-chest images, wrong views, overprocessing, and low-fidelity scans. Our results

---

[2]https://huggingface.co/OpenGVLab/InternVL2-26B

[3]https://huggingface.co/microsoft/rad-dino

confirm that many images in the dataset exhibit these issues. While our approach identifies numerous problematic images, fully curating and removing all low-quality cases is unfeasible due to the substantial human effort required and the absence of well-defined criteria for an automated cleaning pipeline. Furthermore, addressing all instances of low-quality images remains highly challenging through automated processes alone.

**Uncovering Long-tailed Distribution in MIMIC-CXR.** As demonstrated in previous work on natural image-text data (Xu et al., 2023a; Hammoud et al., 2024), a long-tailed distribution in VLP datasets negatively impacts model performance. Therefore, we aim to explore the data distribution of the MIMIC-CXR dataset. However, directly evaluating the text distribution at the sample level, as done in (Xu et al., 2023a), is challenging because each radiology report often describes multiple patterns or anatomical regions, unlike natural image captions that typically focus on a single object (Zhang et al., 2024).

Instead, we adopt an alternative approach by using an off-the-shelf Named Entity Recognition (NER) tool to extract all medical entities, treating them as representatives of the report's concepts and exploring the dataset distribution at the entity level. For this, we use RaTE[4] (Zhao et al., 2024), a model specifically designed for NER tasks on radiology reports. RaTE automatically classifies the extracted entities into five categories: [ABNORMALITY, NON-ABNORMALITY, DISEASE, NON-DISEASE, ANATOMY]. We display the top 50 frequent entiites distribution of each entity type in Fig 2 (c). We display the top 50 frequent entiites distribution of each entity type in Fig 6,7,10,8,9. As shown, all entity types exhibit a severe long-tailed distribution. The MIMIC-CXR (Johnson et al., 2019b) includes a total of 154,049 unique entities, with 55,047 Abnormality, 36,365 Non-Abnormality, 23,017 Disease, 22,103 Non-Disease, and 40,517 Anatomy entities.

### 3.2 Generating Synthetic CXR reports and Paired Images.

Since the MIMIC-CXR dataset (Johnson et al., 2019a) contains various data issues, we generate synthetic radiology reports and CXR images, controlling data quality and distribution during generation to alleviate these problems. In this work, we aim to explore the effectiveness of pretraining MedVLP on a purely synthetic dataset, rather than attempting to create a perfect dataset, as noisy data is unavoidable in real-world scenarios and an ideal dataset is unrealistic.

**CXR Report Generation.** To generate the synthetic reports, the pipeline is depicted in Fig 5. We select a general LLM, Llama3.1-70B-Instruct[5] as the report generator, and we extensively ablate the performance of the report generator with other LLMs in Fig 3. We query the LLM using prompts that include the entity list, as shown in Fig 5.

Since we aim to build a synthetic dataset without a long-tailed distribution, we design a balanced sampling strategy to ensure that the appearance frequency of each entity type is approximately equal across the synthetic dataset. Let $\mathcal{E}$ be the set of entities, categorized into five types: ABNORMALITY, NON-ABNORMALITY, DISEASE, NON-DISEASE, and ANATOMY.

For each generation, we sample:

$$\mathcal{S}_1 = \{e_1^{(i)}, e_2^{(i)}, \ldots, e_k^{(i)}\}, \quad \forall e_j^{(i)} \in \{\text{ABNORMALITY, NON-ABNORMALITY, DISEASE, NON-DISEASE}\}$$

where $k$ is the number of entities sampled from the first four categories. Additionally, we sample:

$$\mathcal{S}_2 = \{a_1^{(i)}, a_2^{(i)}, \ldots, a_m^{(i)}\}, \quad \forall a_j^{(i)} \in \text{ANATOMY}$$

where $m$ is the number of entities sampled from the ANATOMY category. Thus, the total sampled entity set for each generation is:

$$\mathcal{S} = \mathcal{S}_1 \cup \mathcal{S}_2$$

We impose a maximum frequency threshold, $\tau_{\max}$, for each entity $e \in \mathcal{E}$. If an entity $e_j^{(i)}$ in $\mathcal{S}$ reaches this threshold, we resample $e_j^{(i)}$ while keeping the remaining entities in $\mathcal{S}$ unchanged:

$$\text{if } f(e_j^{(i)}) \geq \tau_{\max}, \text{ then resample } e_j^{(i)}.$$

---

[4]https://huggingface.co/Angelakeke/RaTE-NER-Deberta
[5]https://huggingface.co/meta-llama/Meta-Llama-3.1-70B-Instruct

Here, $f(e)$ denotes the current frequency of entity $e$ in the dataset. This ensures a balanced distribution of entities across the synthetic dataset.

After sampling, we input the selected entities $\mathcal{S} = \mathcal{S}_1 \cup \mathcal{S}_2$ into the LLM and indicate their type. Let the output of the LLM be denoted as $R_{gen}$, which represents the synthetic report generated by the model based on the sampled entities. To ensure that the LLM-generated report $R_{gen}$ covers and only includes the entities in $\mathcal{S}$ (since the inclusion of non-specified entities would disrupt the frequency balance), we use the RaTE model (Zhao et al., 2024) to extract entities from $R_{gen}$, denoted as $\mathcal{E}_{gen}$.

We then verify the entity set $\mathcal{E}_{gen}$ by comparing it with the originally sampled set $\mathcal{S}$. If $\mathcal{E}_{gen} \neq \mathcal{S}$, we regenerate the report $R_{gen}$ by repeating the generation process until $\mathcal{E}_{gen} = \mathcal{S}$:

$$\text{if } \mathcal{E}_{gen} \neq \mathcal{S}, \text{ regenerate } R_{gen} \text{ until } \mathcal{E}_{gen} = \mathcal{S}.$$

Once the synthetic report is successfully generated, it is used as the 'FINDINGS' section of the CXR report. We then query the LLM to summarize $R_{gen}$ into the 'IMPRESSION' section, denoted as $R_{imp}$. To ensure consistency between the entities in the 'FINDINGS' and 'IMPRESSION' sections, we extract entities from the summary $R_{imp}$ using RaTE, denoted as $\mathcal{E}_{imp}$. We verify that:

$$\mathcal{E}_{imp} = \mathcal{S}.$$

If the entities in $R_{imp}$ do not match $\mathcal{S}$, we regenerate the "IMPRESSION" section until $\mathcal{E}_{imp} = \mathcal{S}$:

$$\text{if } \mathcal{E}_{imp} \neq \mathcal{S}, \text{ regenerate } R_{imp} \text{ until } \mathcal{E}_{imp} = \mathcal{S}.$$

Given that the number of samples in the original MIMIC-CXR dataset cannot be perfectly divided by $k$ and $m$, we generate a total of 200,000 synthetic samples to ensure a balanced distribution using only off-the-shelf tools, without any specific design for CXR data.

While RadGraph (Delbrouck et al., 2024) could be used for entity extraction, it relies on human-annotated data from MIMIC-CXR and is limited to 16,117 entities. In contrast, RaTE (Zhao et al., 2024) extracts 154,049 entities, making it more suitable for our goal of creating a general and easily transferable pipeline for synthetic data generation. Thus, we chose RaTE for its broader applicability to various radiology reports.

**CXR Image Generation.** After generating the synthetic radiology reports, we aim to generate paired CXR images conditioned on the synthetic reports. Since general text-to-image (T2I) models (e.g., Stable Diffusion) are not designed for CXR image generation and demonstrate poor performance, as shown in (Liu et al., 2023e; Bluethgen et al., 2024), we select RoentGen[6] (Bluethgen et al., 2024), the most recent and validated CXR-specific T2I model, verified by clinicians, as our image generator. We use RoentGen's (Bluethgen et al., 2024) official pretrained weights to generate images. Following their implementation, we use only the 'IMPRESSION' section from the synthetic reports as the text prompt for the T2I model. The generation process is controlled using the official hyperparameters provided by RoentGen, where the classifier-free guidance (CFG) is set to 4 and the number of denoising steps is set to 50.

To prevent the synthetic images from exhibiting the same issues found in the real dataset (as discussed in Sec. 3.1), we apply a similar curation procedure. First, we use the MLLM to filter synthetic images, and then we compute the similarity of visual features between synthetic images and the problematic samples identified from the real dataset. If the visual similarity exceeds a threshold $\delta = 0.5$, we regenerate the images by re-querying the T2I model with the same text prompt until they pass the curation procedure.

We generate 200,000 synthetic CXR images, each paired with a corresponding synthetic report, using only general-purpose, open-source models (e.g., Llama3.1 (AI@Meta, 2024), InternVL2 (Chen et al., 2023)) and vision models pre-trained with self-supervised learning (e.g., RAD-DINO (Pérez-García et al., 2024)). No annotated CXR images or MedVLP models pre-trained on specific CXR image-text datasets are used in this process. This ensures our approach is adaptable and can easily incorporate future advancements in general-purpose models. We refer to this dataset as **SynCXR**.

---

[6]https://stanfordmimi.github.io/RoentGen/

## 3.3 SYNTHETIC DATA TRAINING FOR MEDVLP

Finally, we use the synthetic dataset, SynCXR, to train a MedVLP model and explore how effectively a model can learn from pure synthetic data. Since there are many existing methods for MedVLP, we select simple baseline models like ConVIRT (Zhang et al., 2020) and GLoRIA (Huang et al., 2021) for the following reasons:

**ConVIRT** (Zhang et al., 2020) jointly trains vision and text encoders on paired medical images and reports using global contrastive learning.

**GLoRIA** (Huang et al., 2021) extends ConVIRT by incorporating both global and regional contrastive learning to train the encoders on paired medical images and reports.

These models are open-source, straightforward, and minimize the influence of external factors on evaluating synthetic data for MedVLP. For retraining these two methods on our synthetic dataset, SynCXR, we strictly use their official codebases[7][8]. More complex models may introduce unnecessary complications.

**Excluding Complex Models.** Recent models like BioViL (Boecking et al., 2022) and BioViL-T (Bannur et al., 2023) lack publicly available training code, making them impractical for re-training with synthetic data. Knowledge-enhanced MedVLP models such as MedKLIP, KAD, and MAVL (Wu et al., 2023; Zhang et al., 2023; Phan et al., 2024b) rely on external tools and human-annotated data to incorporate additional knowledge, making direct implementation with synthetic data challenging and introducing unnecessary variables.

## 4 EXPERIMENTS CONFIGURATIONS

For pre-training, we apply the official configurations provided by ConVIRT (Zhang et al., 2020) and GLoRIA (Huang et al., 2021) on the MIMIC-CXR dataset to our synthetic CXR image-text dataset, SynCXR.

### 4.1 DOWNSTREAM TASK DATASETS AND CONFIGURATIONS

For downstream tasks, we evaluate the effectiveness of synthetic data for MedVLP across four tasks. Details on the datasets and implementation are provided in Appendix, Sec A.

**Zero-shot Medical Image Classification.** Following the guidelines in (Phan et al., 2024b; Wu et al., 2023), we perform this task on seven datasets: CheXpert (Saporta et al., 2022), ChestXray-14 (Wang et al., 2017), PadChest-seen, PadChest-unseen, PadChest-rare (Bustos et al., 2020), RSNA (Shih et al., 2019), and SIIM (Steven G. Langer & George Shih, 2019), using the dataset splits from (Phan et al., 2024b). Evaluation metrics include AUC, F1, and ACC.

**Zero-shot Medical Image Visual Grounding.** In line with (Phan et al., 2024b), this task is conducted on the RSNA (Shih et al., 2019), SIIM (Steven G. Langer & George Shih, 2019), and Covid-19 Rural (Desai et al., 2020) datasets, using official splits and metrics. Grounding performance is evaluated with IoU, and Dice score.

**Medical Image Fine-tuned Classification.** As described in (Phan et al., 2024b), we use the RSNA (Shih et al., 2019), SIIM (Steven G. Langer & George Shih, 2019), Covid-19 CXR-2 (Pavlova et al., 2022), and ChestXray-14 (Wang et al., 2017) datasets. During fine-tuning, all model parameters, including the pre-trained vision encoder and linear classifier, are updated. The AdamW optimizer is applied with a learning rate of $1 \times 10^{-4}$, batch size of 64, and training runs for 50 epochs. Evaluation follows the AUC score protocol in (Huang et al., 2021; Wang et al., 2022; Zhou et al.).

**Medical Image Fine-tuned Segmentation.** This task uses the RSNA (Shih et al., 2019), SIIM (Steven G. Langer & George Shih, 2019), and Covid-19 Rural (Desai et al., 2020) datasets, following preprocessing from (Wang et al., 2022; Huang et al., 2021). U-Net (Ronneberger et al., 2015) is used for fine-tuning, freezing the pre-trained vision encoder and updating only the decoder parameters.

---

[7]https://github.com/marshuang80/gloria
[8]https://github.com/edreisMD/ConVIRT-pytorch

| Method | Pre-training Data | CheXpert | | ChestXray-14 | | PadChest-seen | | RSNA | | SIIM | |
|--------|-------------------|------|------|------|------|------|------|------|------|------|------|
| | | AUC ↑ | F1 ↑ | AUC ↑ | F1 ↑ | AUC ↑ | F1 ↑ | AUC ↑ | F1 ↑ | AUC ↑ | F1 ↑ |
| ConVIRT | MIMIC-CXR | 52.10 | 35.61 | 53.15 | 12.38 | 63.72 | 14.56 | 79.21 | 55.67 | 64.25 | 42.87 |
| | SynCXR | 59.49 | 40.51 | 56.07 | 15.43 | 63.43 | 15.10 | 82.08 | 58.38 | 75.55 | 57.43 |
| | **Mix** | **71.54** | **47.11** | **61.28** | **18.52** | **68.48** | **16.67** | **83.86** | **61.28** | **78.51** | **59.10** |
| GLoRIA | MIMIC-CXR | 54.84 | 37.86 | 55.92 | 14.20 | 64.09 | 14.83 | 70.37 | 48.19 | 54.71 | 40.39 |
| | SynCXR | 61.38 | 41.05 | 57.47 | 15.60 | 64.26 | 15.02 | 72.34 | 49.50 | 67.32 | 53.86 |
| | **Mix** | **72.32** | **48.54** | **61.06** | **17.33** | **68.35** | **17.00** | **74.32** | **51.10** | **73.49** | **56.09** |

Table 1: Performance of zero-shot classification on five datasets for diseases present in the MIMIC-CXR dataset, evaluated on two MedVLP models pretrained on MIMIC-CXR (real) and SynCXR (**pure synthetic**). 'Mix' denotes the direct combination of real and synthetic data for MedVLP pretraining. Best results are highlighted in bold.

| Method | Pre-training Data | Covid-19 CXR-2 | | PadChest-unseen | | PadChest-rare | |
|--------|-------------------|------|------|------|------|------|------|
| | | AUC ↑ | F1 ↑ | AUC ↑ | F1 ↑ | AUC ↑ | F1 ↑ |
| ConVIRT | MIMIC-CXR | 62.78 | 71.23 | 51.17 | 4.12 | 50.37 | 3.31 |
| | SynCXR | 64.41 | 72.03 | 54.47 | 4.51 | 53.70 | 3.69 |
| | Mix | **69.23** | **72.85** | **58.53** | **5.35** | **57.68** | **4.40** |
| GLoRIA | MIMIC-CXR | 64.52 | 70.78 | 49.96 | 4.07 | 48.25 | 3.41 |
| | SynCXR | 66.70 | 71.90 | 54.24 | 4.10 | 51.26 | 3.75 |
| | **Mix** | **68.76** | **73.22** | **58.60** | **5.60** | **58.58** | **4.62** |

| Method | Pre-training Data | RSNA | | Covid-19 Rural | | SIIM | |
|--------|-------------------|------|------|------|------|------|------|
| | | IoU ↑ | Dice ↑ | IoU ↑ | Dice ↑ | IoU ↑ | Dice ↑ |
| ConVIRT | MIMIC-CXR | 18.93 | 28.45 | 7.42 | 10.55 | 3.01 | 8.74 |
| | SynCXR | 22.98 | 31.45 | 8.62 | 10.83 | 3.43 | 9.67 |
| | **Mix** | **25.97** | **34.25** | **12.78** | **14.12** | **4.58** | **11.43** |
| GLoRIA | MIMIC-CXR | 21.82 | 34.68 | 8.18 | 12.49 | 3.11 | 10.23 |
| | SynCXR | 23.00 | 35.25 | 9.47 | 13.00 | 3.50 | 10.75 |
| | Mix | **26.34** | **36.52** | **12.67** | **14.63** | **4.51** | **11.73** |

(a) Performance of zero-shot classification on three datasets for unseen diseases.

(b) Performance of zero-shot grounding on RSNA, SIIM, and Covid-19 Rural.

Table 2: Zero-shot tasks performance of MedVLP models on disease classification (a) and grounding (b) across multiple datasets, using MIMIC-CXR, SynCXR, and Mix datasets for pretraining.

Performance is measured using the Dice score, adhering to the evaluation protocol from (Huang et al., 2021).

## 4.2 EXPERIMENTAL RESULTS

Since the MIMIC-CXR dataset already includes several diseases present in downstream tasks, as mentioned in (Phan et al., 2024b; Zhang et al., 2023), we split the zero-shot classification task into *seen* and *unseen* categories, strictly following (Phan et al., 2024b). Note that all experimental results for ConVIRT and GLoRIA pre-trained with real data (MIMIC-CXR) are directly referenced from (Phan et al., 2024b) to ensure a fair comparison.

**Zero-shot Classification on Seen Diseases.** Tab 1 shows the zero-shot classification performance on *seen* diseases. Across all datasets, both MedVLP methods pretrained on SynCXR (our **purely synthetic dataset**) consistently outperform or achieve comparable performance to their counterparts pretrained on real datasets, with an average improvement of 4.7% in AUC and 4.53% in F1 scores. Furthermore, the methods pretrained on the mixed dataset, which directly combines real and synthetic data, achieve even greater improvements, with 10.08% AUC and 7.62% F1 scores on average across all datasets and methods. This demonstrates that the SynCXR dataset effectively enables MedVLP models to learn representative cross-modal features, enhancing their zero-shot classification capability.

**Zero-shot Classification on Unseen Diseases.** Tab 2a reports the zero-shot classification performance on *unseen* diseases. Similar to the results for seen diseases, MedVLP models pretrained on the synthetic dataset consistently outperform those pretrained on real data, with an average improvement of 2.96% AUC and 0.51% F1 scores. Additionally, models pretrained on the mixed dataset show substantial gains over those trained on real data, with 7.39% AUC and 1.52% F1 scores on average. This indicates that the SynCXR dataset, generated with meticulous quality control and balanced distribution, can increase the generalizability of MedVLP models for unseen diseases prediction.

**Zero-shot Visual Grounding.** We further evaluate the effectiveness of synthetic data in improving MedVLP models' local visual understanding capabilities through zero-shot grounding tasks. Tab 2b presents the performance of zero-shot grounding on RSNA (Shih et al., 2019), Covid-19 Rural (Desai et al., 2020), and SIIM (Steven G. Langer & George Shih, 2019). Across all datasets, MedVLP models pretrained on the SynCXR dataset achieve superior performance compared to those trained on the real dataset, with an average increase of 1.42% IoU and 0.97% Dice scores. The mixed dataset further enhances performance, with 4.06% IoU and 2.92% Dice scores on average. This demonstrates that the SynCXR dataset not only benefits global cross-modal feature learning but also improves local visual understanding for MedVLP models.

| Task | Classification | | | | | | | | | | | | Segmentation | | | | | | | | |
|---|---|---|---|---|---|---|---|---|---|---|---|---|---|---|---|---|---|---|---|---|---|
| Dataset | RSNA | | | SIIM | | | Covid19 CXR-2 | | | ChestXray-14 | | | RSNA | | | Covid-19 Rural | | | SIIM | | |
| Data Ratio | 1% | 10% | 100% | 1% | 10% | 100% | 1% | 10% | 100% | 1% | 10% | 100% | 1% | 10% | 100% | 1% | 10% | 100% | 1% | 10% | 100% |
| ConVIRT-Real | 78.86 | 85.42 | 87.64 | 72.39 | 80.41 | 91.67 | 90.30 | 97.74 | 99.70 | 57.23 | 72.53 | 79.13 | 56.48 | 63.94 | 71.87 | 16.97 | 30.79 | 42.71 | 28.75 | 47.21 | 65.75 |
| ConVIRT-Syn | 79.01 | 85.58 | 87.90 | 73.51 | 81.10 | 91.84 | 91.50 | 98.80 | 99.73 | 57.45 | 73.60 | 80.20 | 58.00 | 65.10 | 72.90 | 17.10 | 32.00 | 43.90 | 29.90 | 48.50 | 66.81 |
| **ConVIRT-Mix** | **79.75** | **86.21** | **88.45** | **73.00** | **82.80** | **92.31** | **91.81** | **99.00** | **99.81** | **57.61** | **74.20** | **80.51** | **58.50** | **65.81** | **73.30** | **18.40** | **32.50** | **44.21** | **30.10** | **48.81** | **67.11** |
| GLoRIA-Real | 79.13 | 85.59 | 87.83 | 75.85 | 86.20 | 91.89 | 92.74 | 97.18 | 99.54 | 58.94 | 72.87 | 79.92 | 58.13 | 67.71 | 72.06 | 16.12 | 31.20 | 43.85 | 31.87 | 40.61 | 64.82 |
| GLoRIA-Syn | 80.30 | 86.75 | 88.00 | 76.01 | 87.40 | 92.11 | 94.01 | 98.41 | 99.75 | 60.11 | 74.01 | 81.11 | 60.41 | 70.01 | 73.51 | 17.31 | 32.51 | 45.01 | 32.91 | 41.91 | 66.01 |
| **GLoRIA-Mix** | **81.01** | **87.50** | **88.61** | **77.51** | **88.01** | **92.51** | **94.51** | **99.61** | **99.86** | **60.31** | **74.51** | **81.51** | **61.01** | **70.51** | **74.01** | **17.51** | **33.01** | **45.31** | **33.51** | **42.21** | **67.51** |

Table 3: Results from two MedVLP methods pre-trained on real, synthetic, and mixed datasets are reported for classification (AUC) and segmentation (Dice) tasks. 'ConVIRT-Real' and 'GLoRIA-Real' refer to models pre-trained on MIMIC-CXR using real data, while 'ConVIRT-Syn' and 'GLoRIA-Syn' indicate models pre-trained on SynCXR using synthetic data. 'ConVIRT-Mix' and 'GLoRIA-Mix' represent models trained on a combination of MIMIC-CXR and SynCXR. Best results are in bold.

**Fine-tuning Tasks.** To evaluate the representation quality learned by MedVLP, we report the fine-tuned classification and segmentation performance in Tab 3. Similar to the zero-shot task, MedVLP models pre-trained on SynCXR consistently outperform those trained on the real dataset across all data ratios for both classification and segmentation tasks. Furthermore, the combination of real and synthetic datasets (Mix) further boosts performance, demonstrating that SynCXR data not only enhances cross-modal representation learning but also improves performance in single-modal tasks.

# 5 ANALYSIS

| Method | Entity Sampling Strategy | Avg. Zero-shot Classification |
|---|---|---|
| ConVIRT (Zhang et al., 2020) | **w/ balance Sampling** | **63.65** |
| | w/o balance Sampling | 60.21 |
| GLoRIA (Huang et al., 2021) | **w/ balance Sampling** | **61.87** |
| | w/o balance Sampling | 58.42 |

(a) Impact of Entity Sampling Strategies

| Method | Real Image | Syn. Image | Real Report | Syn. Report | Avg. Zero-shot Classification |
|---|---|---|---|---|---|
| ConVIRT (Zhang et al., 2020) | ✓ | | ✓ | | 59.59 |
| | | ✓ | ✓ | | 61.04 |
| | ✓ | | | ✓ | 59.36 |
| | | ✓ | | ✓ | **63.65** |
| GLoRIA (Huang et al., 2021) | ✓ | | ✓ | | 57.83 |
| | | ✓ | ✓ | | 58.62 |
| | ✓ | | | ✓ | 57.69 |
| | | ✓ | | ✓ | **61.87** |

(b) Impact of Different Synthetic Data

Table 4: Evaluation of entity sampling strategies for synthetic report generation and the impact of synthetic data types on MedVLP.

**Effect of Balanced Entity Sampling in Generating Synthetic Reports.** We evaluate the impact of balanced sampling entities when generating synthetic reports using LLMs. For the synthetic dataset without balanced sampling, we adjust entity frequencies to match their distribution in MIMIC-CXR, leading to a long-tailed distribution. As shown in Tab 4a, for both MedVLP methods, the performance improves significantly when using synthetic datasets generated from balanced sampled entities. This demonstrates that balanced sampling of entities leads to a more representative dataset, benefiting MedVLP performance.

**Evaluating the Contribution of Synthetic Images and Reports.** We aim to assess the individual impact of synthetic images and synthetic reports on MedVLP performance. As shown in Tab 4b, we generate two partially synthetic datasets by replacing either the image or the text with synthetic data, while keeping the other components real, to evaluate their respective contributions.

- **Real Image, Synthetic Report:** In this setting, we use MedVersa[9] (Zhou et al., 2024), a state-of-the-art radiology report generation model, to generate synthetic reports for each real CXR image. We then train MedVLP models using these real image and synthetic report pairs.
- **Real Report, Synthetic Image:** In this setting, we use RoentGen (Bluethgen et al., 2024), a text-to-image model, to generate synthetic CXR images for each real report. The 'IMPRESSION' section of each report serves as the prompt for generating synthetic CXR images. These synthetic image and real report pairs are used to train MedVLP models.

According to Tab 4b, for both MedVLP methods, using real images with synthetic reports results in decreased performance, likely due to the persistent long-tailed distribution, as the synthetic reports are generated based on real images. However, using real reports with synthetic images slightly improves performance, as synthetic images can be curated using our image filtering procedure to ensure high quality, avoiding issues commonly found in real datasets. Using both synthetic images and synthetic reports achieves the highest performance, indicating that a well-curated synthetic dataset can significantly enhance MedVLP performance.

---

[9]https://huggingface.co/hyzhou/MedVersa

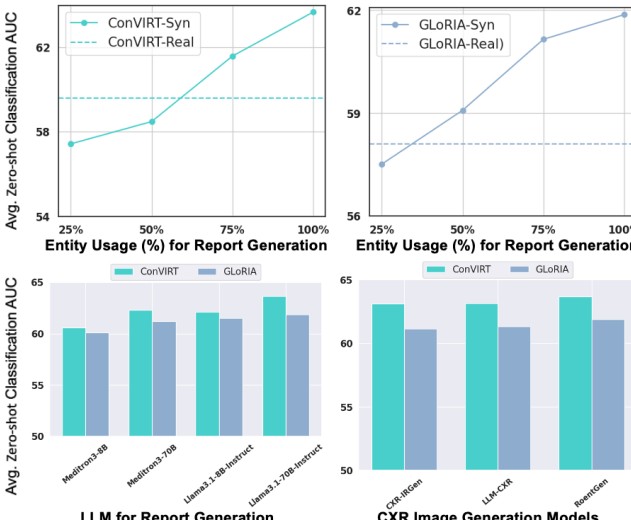

Figure 3: Effectiveness of various factors on SynCXR dataset. **Top:** Impact of entity usage ratio on MedVLP performance for ConVIRT and GLoRIA methods. **Bottom Left:** Effectiveness of different LLMs for report generation on both MedVLP methods. **Bottom Right:** Effectiveness of different CXR image generation models for both MedVLP methods.

**Impact of Entity Diversity.** We evaluate the impact of entity diversity by varying the number of entities used for generating the SynCXR dataset. We generate synthetic datasets using 25%, 50%, and 75% of these entities, following the same procedure each time. The results, shown in Fig 3 (Top), indicate that zero-shot classification performance improves as more entities are used for report generation. This suggests that increasing dataset diversity positively influences downstream performance.

**Impact of Different Report Generators.** We also examine the impact of using different LLMs for synthetic report generation. As shown in Fig 3 (Bottom Left), we compare two general LLMs, LLaMA 3.1 (8B and 70B), and two medical-specific LLMs, Meditron3 (8B[10] and 70B[11]). Despite Meditron3 being trained specifically on medical corpora and inheriting weights from LLaMA, the dataset generated by LLaMA 3.1-70B-Instruct achieves the best performance. This indicates that a powerful general LLM is effective for generating synthetic datasets, and using domain-specific fine-tuned versions may degrade the quality of the synthetic data.

**Impact of Different Image Generators.** We evaluate various text-to-image models for synthetic CXR image generation, including CXR-IRGen (Shentu & Al Moubayed, 2024), LLM-CXR (Lee et al., 2023), and RoentGen (Bluethgen et al., 2024). As shown in Fig 3 (Bottom Right), datasets generated by RoentGen lead to the best performance for both MedVLP methods. This is likely because RoentGen is the only image generation model verified by clinicians, suggesting that the quality of image generation models is crucial for building synthetic datasets, and models should be validated by clinical experts.

## 6 CONCLUSION

In this work, we tackle the question: ***Can MedVLP succeed using purely synthetic data?*** Our findings demonstrate that the answer is: ***Yes***. To the best of our knowledge, this is the first study to comprehensively explore the potential of synthetic data for MedVLP models. We also identify key limitations in existing real-world datasets and introduce SynCXR—a synthetic dataset of 200,000 image-text pairs generated without any manual quality checks. Our findings show that MedVLP models trained on purely synthetic data outperform those trained on real data. Moreover, combining synthetic and real data further boosts model performance, demonstrating the potential of synthetic data to overcome limitations in real-world datasets. We systematically analyze key factors in SynCXR and validate its effectiveness through extensive ablation studies. In summary, we show that MedVLP achieves strong performance using a purely synthetic image-text dataset and benefits significantly from a combination of real and synthetic data. We believe this work will inspire the community to fully leverage synthetic data and mitigate the challenges posed by noisy and limited real-world datasets.

---

[10] https://huggingface.co/OpenMeditron/Meditron3-8B
[11] https://huggingface.co/OpenMeditron/Meditron3-70B

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

## A  DOWNSTREAM TASKS CONFIGURATION

### A.1  DATASET DETAILS

In this section, we provide details on all datasets used. The dataset splits are publicly accessible at[12].

**ChestX-ray14** (Wang et al., 2017) includes 112,120 frontal X-rays from 30,805 patients, labeled for 14 diseases. We use the official split and partition it into 80%/10%/10% for train/validation/test.

**PadChest** (Bustos et al., 2020) includes 160,868 X-rays from 67,000 patients, annotated with over 150 findings. As in (Phan et al., 2024b), three subsets are built based on PadChest: 14 common diseases as **PadChest-seen**, rare diseases from the NORD database[13] as **PadChest-rare**, and the remaining diseases as **PadChest-unseen**. We use the official split provided by (Phan et al., 2024b).

**RSNA** (Shih et al., 2019) contains over 260,000 frontal X-rays annotated with pneumonia masks. We divide it into training (60%), validation (20%), and test (20%) sets for segmentation and classification tasks (Huang et al., 2021; Wu et al., 2023).

**CheXpert** (Irvin et al., 2019) contains 224,316 chest X-rays from 65,240 patients at Stanford Hospital, with an official validation set of 200 studies and a test set of 500 studies, both annotated by board-certified radiologists. Our evaluation on the five observations in the official test set follows protocols from earlier studies (Tiu et al., 2022b; Irvin et al., 2019).

**SIIM** (Steven G. Langer & George Shih, 2019) consists of over 12,000 frontal X-rays annotated with pneumothorax masks, split into training (60%), validation (20%), and test (20%) sets.

**COVIDx CXR-2** (Wang et al., 2020) includes 29,986 X-rays from 16,648 COVID-19 patients, divided into training (70%), validation (20%), and test (10%) (Pavlova et al., 2022).

**COVID Rural** (Desai et al., 2020) contains over 200 X-rays with segmentation masks, divided into training (60%), validation (20%), and test (20%).

### A.2  IMPLEMENTATION DETAILS

**Zero-shot Image Classification.** The CXR images undergo a two-step preprocessing: resizing to $256 \times 256$, followed by center cropping to $224 \times 224$. As per (Huang et al., 2021), pixel values are normalized to $[0, 1]$. The processed image is passed through a visual encoder and projector to generate the image embedding $\hat{\mathbf{v}}_i$. Simultaneously, the text prompts are processed through a text encoder to obtain text embeddings $\hat{\mathbf{l}}_i$. Classification is based on cosine similarity between image and text embeddings. If the similarity between the image embedding and the positive prompt (e.g., _disease_) is higher than that with the negative prompt (e.g., _No disease_), the classification is positive, and vice versa. The prompt design follows (Tiu et al., 2022a) for both ConVIRT and GLoRIA.

**Zero-shot Visual Grounding.** For this task, we follow the BioViL pipeline as described in (Phan et al., 2024b), since ConVIRT (Zhang et al., 2020) and GLoRIA (Huang et al., 2021) do not provide code for visual grounding. This pixel-level classification task relies on the similarity between text embeddings and the dense visual feature map from the final convolutional layer. The cosine similarity generates a similarity map, resized to match the image, and used as segmentation results for grounding evaluation.

**Medical Image Fine-tuned Classification.**

For fine-tuning, we follow the experimental setup from (Phan et al., 2024b), updating both the visual encoder and linear layer. Images are resized to $256 \times 256$, and data augmentation is applied as recommended in (Zhang et al., 2023). We use the AdamW optimizer with a learning rate of $1 \times 10^{-4}$, batch size of 64, for 50 epochs on a single A100 GPU. Early stopping is applied, with a learning rate of 5e-4 and batch size of 8. AdamW is configured with $\beta_1 = 0.9$, $\beta_2 = 0.999$, and weight decay of 1e-6.

**Medical Image Fine-tuned Segmentation.** For segmentation tasks on the RSNA (Shih et al., 2019), SIIM (Steven G. Langer & George Shih, 2019), and Covid-19 Rural (Wang et al., 2020) datasets, we

---

[12]https://github.com/HieuPhan33/CVPR2024_MAVL/tree/main/data

[13]https://rarediseases.org/rare-diseases/

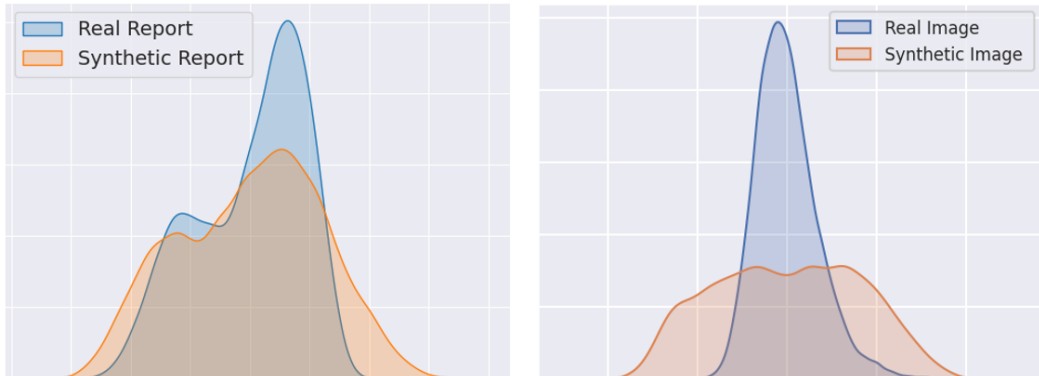

Figure 4: Distribution of Synthetic and Real Data. (a): Comparison of the first principal component distribution of features extracted from RAD-DINO for synthetic and real images. (b): Comparison of the first principal component distribution of features extracted from Med-CPT for synthetic and real reports.

fine-tune both the pre-trained vision encoder and decoder. Training is performed with early stopping at 50 epochs, using a learning rate of 2e-4 and weight decay of 0.05. AdamW is the optimizer, with $\beta_1 = 0.9$ and $\beta_2 = 0.999$. Batch sizes are 8 for SIIM and 16 for RSNA. All configurations follow the protocol from (Huang et al., 2021).

## B    EXTRA VISUALIZATION

**Distribution of Synthetic and Real Data.** We illustrate the distribution of synthetic and real data in Fig 4. For visualization, we use RAD-DINO (Pérez-García et al., 2024) to extract image features and Med-CPT (Jin et al., 2023) to extract report features. We then apply Principal component analysis (PCA) to reduce the feature dimensions and visualize the first principal component. As shown in Fig 4, the synthetic data covers a broader range than the real data, indicating greater diversity. In contrast, the real data shows a more concentrated distribution, which may limit the generalizability of MedVLP models.

**Pipeline of Synthetic Report Generation.** The pipeline for generating synthetic reports using LLMs and balanced sampled clinical entities is illustrated in Fig 5.

**Entities Distribution.** We visualize the distribution of each type of entity in the MIMIC-CXR dataset. Due to space constraints, only the top 200 most frequent entities are shown, revealing a clear long-tailed distribution in Fig 6, 10, 8, 7, and 9.

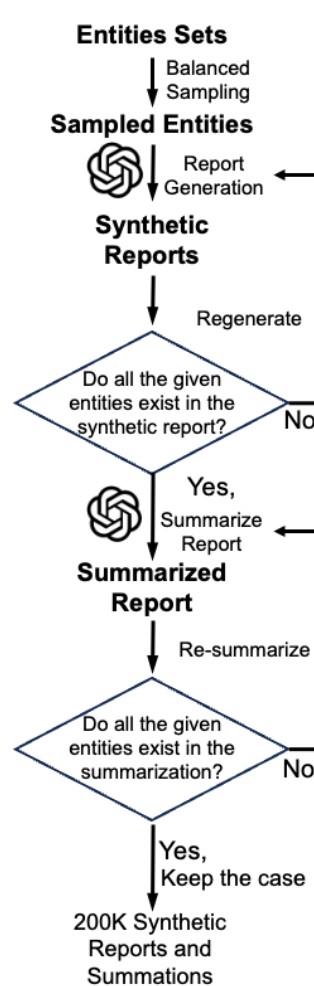

Figure 5: Pipeline for generating synthetic reports. The process begins by generating the 'FINDINGS' section, followed by summarizing it into the 'IMPRESSION' section. Both sections are checked to ensure they contain the specified entities; if not, the generation process is repeated. The final dataset includes 200,000 synthetic reports, each containing both 'FINDINGS' and 'IMPRESSION' sections.

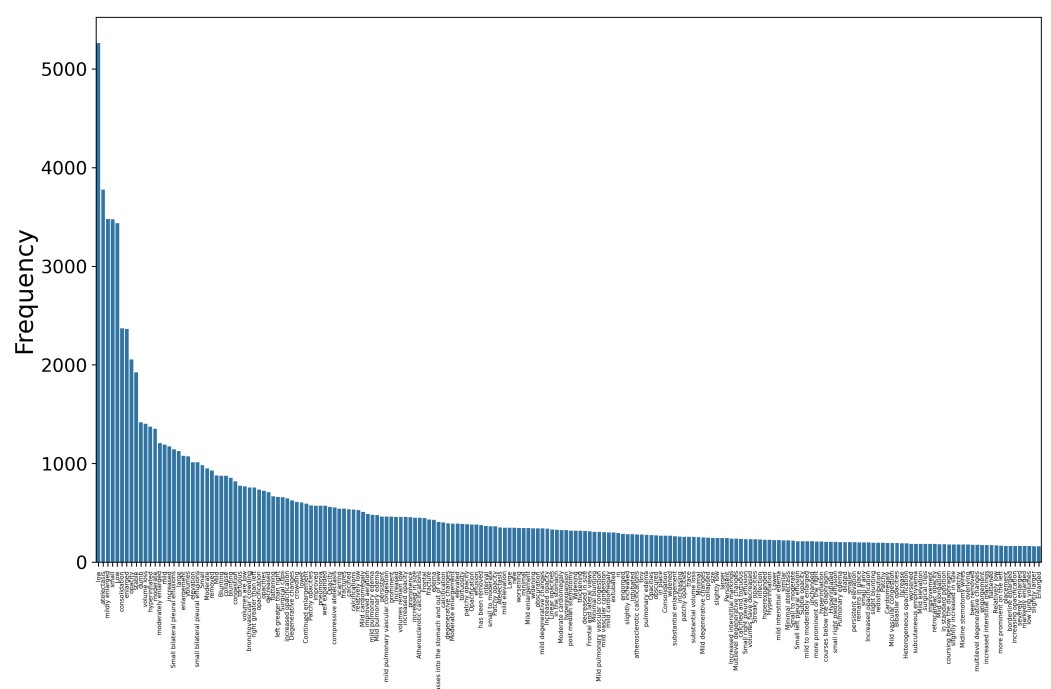

Figure 6: Top 200 most frequent abnormality entities.

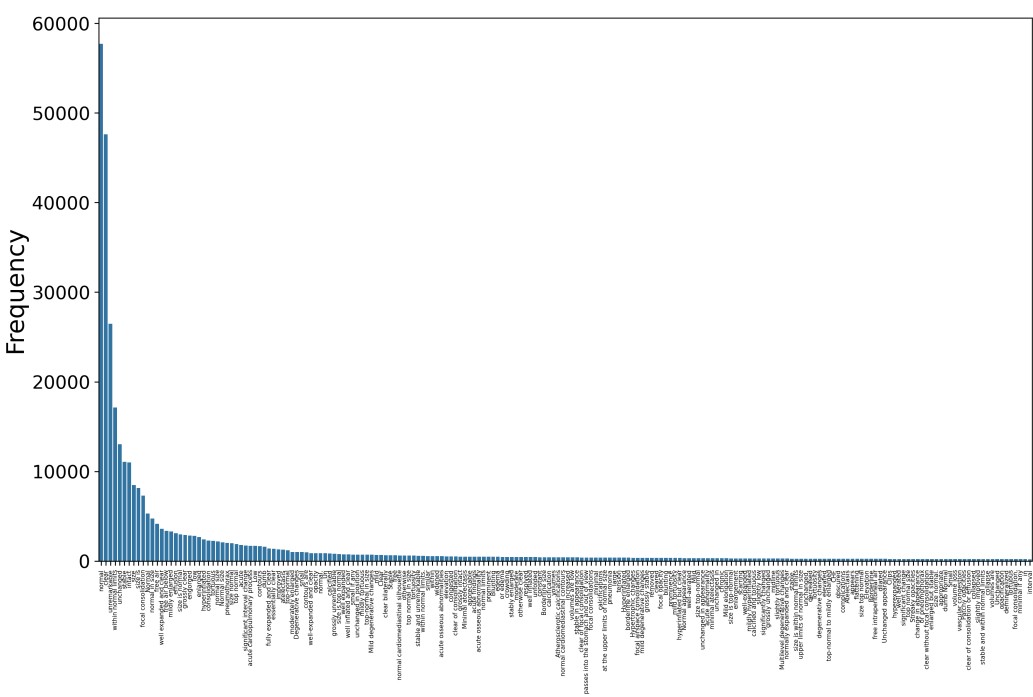

Figure 7: Top 200 most frequent non-abnormality entities.

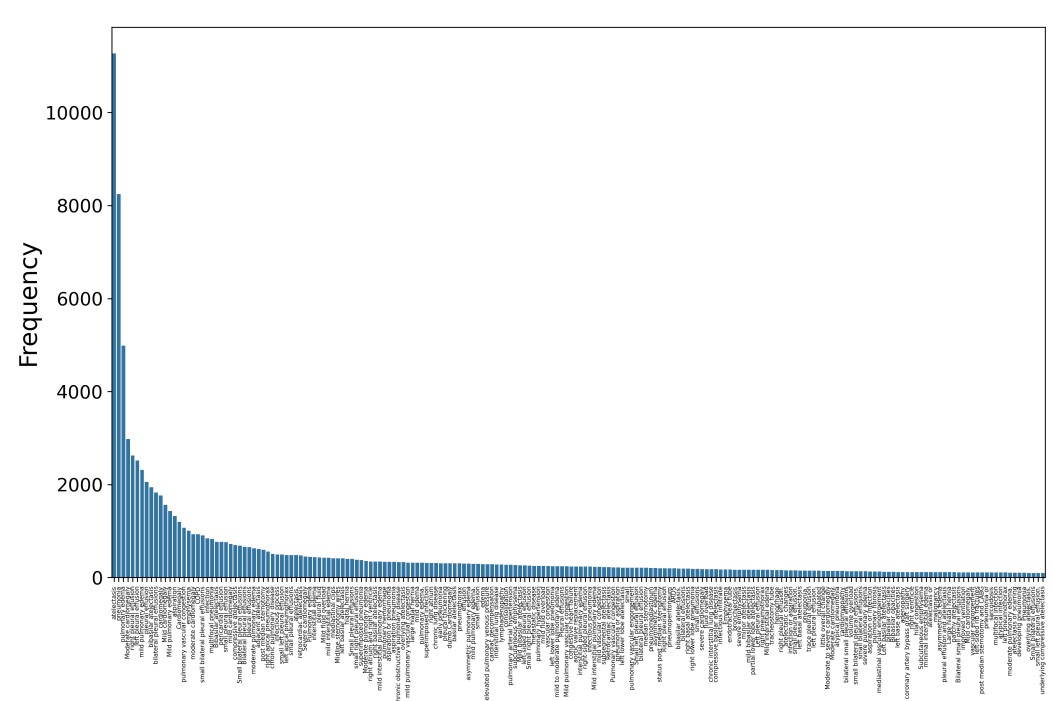

Figure 8: Top 200 most frequent disease entities.

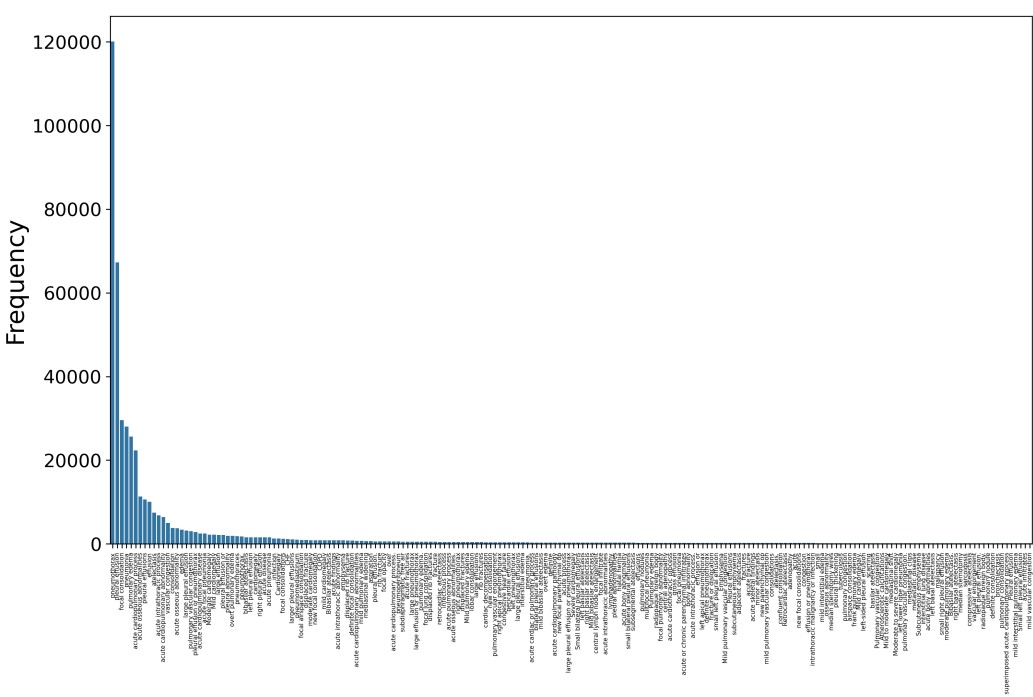

Figure 9: Top 200 most frequent non-disease entities.

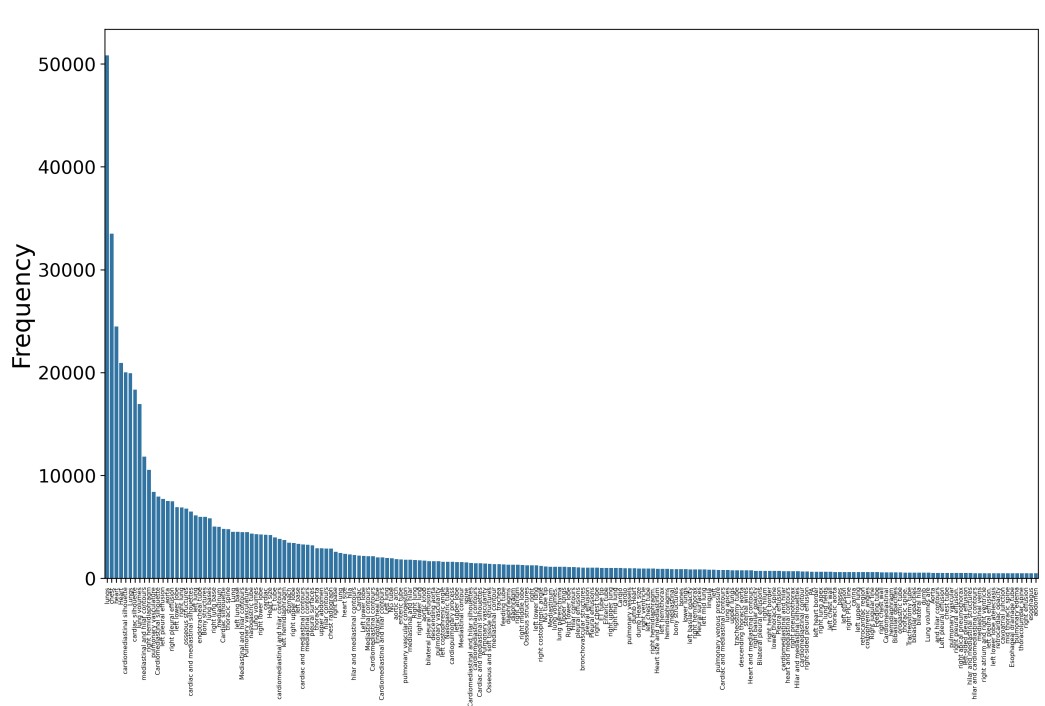

Figure 10: Top 200 most frequent anatomy entities.

