# OpenReview forum: "Can Medical Vision-Language Pre-training Succeed with Purely Synthetic Data?"
_ICLR.cc/2025/Conference — ICLR 2025 Conference Withdrawn Submission_

### Official Review · Reviewer_34ZS · 2024-10-31

**Soundness:** 2
**Presentation:** 3
**Contribution:** 2
**Rating:** 5
**Confidence:** 5

**Summary:**

Given that recent large visual language models require extensive medical data and often rely on synthetic data, the authors question whether synthetic data alone could meet these needs effectively. Through experiments conducted on a synthetic chest X-ray dataset, they found that the synthetic data outperformed real data.

**Strengths:**

- The authors developed SYNCXR, a large-scale synthetic X-ray dataset, demonstrating that models trained on this dataset could achieve improved performance.
- They also identified common issues prevalent in real-world datasets, such as MIMIC-CXR.

**Weaknesses:**

- Concerns with the Experiment and Scope: The focus of this study should be limited to evaluating whether a synthetic CXR dataset can enhance VLM training. Currently, the authors have not included experiments involving other modalities, which limits the broader applicability of the findings.
- RadGraph and RaTE Discussion: I recommend addressing the limitations of RadGraph and the rationale behind RaTE in the related work section, rather than in the main text, as it detracts from the primary focus.
- Novelty and Methodological Concerns: This work primarily takes an engineering approach to address a specific task, lacking novel theoretical contributions or robust validation. For instance, the authors did not provide a mathematical explanation as to whether synthetic data can lead to improved representation learning, nor did they validate the generated data with real clinicians. While model-based validation is acceptable, practical relevance in radiology requires clinician verification.
- For medical image segmentation, the authors should include qualitative comparisons across methods to strengthen their claims, especially as the current quantitative results are presented as single-digit figures, which may be insufficient to persuade readers.
- Additional experiments are recommended to expand the study’s scope, including evaluations across other modalities, tasks beyond visual language pretraining, and models that confidently showcase quantitative results.
- There is a typo in line 233.

**Questions:**

- The authors noted that 1,448 + 5,512 samples were removed from the MIMIC-CXR dataset; however, this constitutes only about 3% of the entire dataset, raising questions about the necessity of this step. Could the authors provide evidence demonstrating that excluding these ~7,000 samples significantly benefits MedVLP training?
- Incorporating more modalities and tasks would strengthen the argument that synthetic datasets can offer substantial benefits across a broader range of applications.

---

> ### Author Response · Authors · 2024-11-23
>
> > **W1, Q2:** Experiments on Other Modalities
> - Currently, there is no publicly accessible generative model, verified by clinicians, that can generate 3D CT or MRI volumes from text conditions. We are aware of recent work [1] that explores text-conditional CT generation. However, the quality of the generated images is still limited and has not been validated by clinicians. As a result, this model is not suitable for training a MedVLP model using purely synthetic data.
>
>   In contrast, RoentGen has been validated by clinicians and has demonstrated satisfactory performance in studies evaluated by human experts [2].
>
> [1] Hamamci, Ibrahim Ethem, et al. "GenerateCT: text-conditional generation of 3D chest CT volumes." European Conference on Computer Vision. Springer, Cham, 2025.
> [2] Bluethgen, Christian, et al. "A vision–language foundation model for the generation of realistic chest x-ray images." *Nature Biomedical Engineering* (2024): 1-13.

---

> ### Author Response · Authors · 2024-11-23
>
> > **W2:** Moving RadGraph and RaTE to the Related Work
> - Thank you for your detailed advice. We will move this content to the Related Work section (Section 2) in the camera-ready version.
>
> > **W3:** Without Theoretical Explanation or Clinician Verification
>
> - First, we kindly disagree with the claim that our work is solely an engineering approach for a specific task. Our work systematically explores and addresses key issues in existing MedVLP datasets, such as "unpaired image-text samples," "low-quality images," and "long-tailed distributions in text." Using our proposed pipeline, we identify these challenges and design a generation framework to mitigate them, rather than brute-forcing the generation of synthetic samples. Additionally, we evaluate models pre-trained on our synthetic data across four downstream tasks and nine datasets, demonstrating the effectiveness of our approach.
>
> - Our study investigates whether VLP can succeed using purely synthetic data and whether VLP models trained on such data can learn general representations that benefit a wide range of downstream tasks. While we acknowledge that synthetic images may contain inaccuracies, they are utilized solely during the vision-language pretraining stage to develop general representations. At this stage, the presence of noisy data is not unusual. For instance, the CLIP-LAION2B dataset includes noisy samples, and the MIMIC-CXR dataset, as demonstrated in our work and noted in BioViL-T [1], contains numerous unpaired image-text samples. Despite these imperfections, VLP methods pre-trained on these datasets have consistently demonstrated strong performance on downstream tasks across both natural and medical imaging domains [2,3]. This indicates that Vision-Language Pretraining models are capable of learning robust representative features even in the presence of noisy or imperfectly paired data.
>
> - Therefore, while clinician verification is essential for clinical deployment, thoroughly validating synthetic samples for the VLP stage is not strictly necessary.
>
>
> [1] Bannur, Shruthi, et al. "Learning to exploit temporal structure for biomedical vision-language processing." Proceedings of the IEEE/CVF Conference on Computer Vision and Pattern Recognition. 2023.
>
> [2] Wang, Fuying, et al. "Multi-granularity cross-modal alignment for generalized medical visual representation learning." Advances in Neural Information Processing Systems 35 (2022): 33536-33549.
>
> [3] Radford, Alec, et al. "Learning transferable visual models from natural language supervision." International conference on machine learning. PMLR, 2021.
>
> > **W4:** Qualitative Results of the Segmentation Task
> - Thank you for your feedback. We will update the visualization results of the segmentation task in the camera-ready version.
>
> > **W5:** Experiment on Other Modalities
> - Our work focuses on evaluating the effectiveness of purely synthetic data for VLP, which relies heavily on a clinician-verified image generative model. Without such verification, the quality of synthetic data would be inadequate. Currently, there is no publicly accessible generative model, validated by clinicians, that can generate 3D CT or MRI volumes from text conditions.
>
>   We are aware of recent work [1] that explores text-conditional CT generation. However, the quality of the generated images remains limited and has not been validated by clinicians, making it unsuitable for training a MedVLP model using purely synthetic data.
>
>   In contrast, RoentGen, designed for chest X-ray image generation, has been validated by clinicians and has demonstrated satisfactory performance in studies evaluated by human experts [2]. Hence, this work focuses on synthetic chest X-ray image-text pairs. Nevertheless, our method can be directly extended to other modalities as soon as high-quality generative models become available for those modalities, since our approach does not rely on any modality-specific priors.
>
> [1] Hamamci, Ibrahim Ethem, et al. "GenerateCT: text-conditional generation of 3D chest CT volumes." *European Conference on Computer Vision.* Springer, Cham, 2025.
> [2] Bluethgen, Christian, et al. "A vision–language foundation model for the generation of realistic chest x-ray images." *Nature Biomedical Engineering* (2024): 1-13.
>
> > **W6:** Typos in Line 233
> - Thank you for pointing this out. We will correct *top 50 frequent entiites* to *top 50 frequent entities* in the camera-ready version.

---

> ### Author Response · Authors · 2024-11-23
>
> > **Q1:** Results on Cleaned MIMIC-CXR Excluding 7,000+ Samples
>
> - First, we mention these 7,000+ samples in our work to demonstrate that we successfully identified data issues in the real dataset, not to suggest that the dataset can be perfectly cleaned by removing these samples. Achieving a completely clean dataset by eliminating every problematic sample is impractical.
>
> - Therefore, we do not expect the filtering procedure to significantly enhance VLP performance. While we removed bad samples based on image quality, another critical issue we identified—the long-tailed distribution of entities in the text—cannot be resolved by simply removing samples. This is because each report may contain entities from both head and tail groups. For example, one report could include entities that are both frequent and rare, so removing the report would not effectively address the long-tailed distribution. Consequently, even after filtering, the real dataset still suffers from this issue, which negatively affects VLP performance.
>
> - We re-pretrained ConVIRT on the cleaned version of MIMIC-CXR, excluding these 7,000+ samples, and the results are shown below. As demonstrated in the table, the VLP method pre-trained on the filtered MIMIC-CXR dataset slightly outperforms the version pre-trained on the uncleaned dataset. However, it still underperforms compared to the method pre-trained on our synthetic data.
>
> This can be attributed to two key factors:
>
> 1. Even after removing many imperfect samples, it remains impractical to eliminate all poor or unpaired samples from the real dataset. Consequently, some problematic samples persist, negatively impacting the VLP process.
>
> 2. The cleaning process primarily focused on filtering samples based on image quality. However, the long-tailed distribution of entities within the dataset was not addressed. Since image-text pairs cannot be easily downsampled or oversampled, this issue limits the representational balance and continues to affect the cleaned dataset.
>
>
> | Method  | Pre-training Data       | CheXpert (AUC ↑ / F1 ↑) | ChestXray-14 (AUC ↑ / F1 ↑) | PadChest-seen (AUC ↑ / F1 ↑) | RSNA (AUC ↑ / F1 ↑) | SIIM (AUC ↑ / F1 ↑) |
> |---------|-------------------------|--------------------------|-----------------------------|------------------------------|---------------------|---------------------|
> | ConVIRT | MIMIC-CXR               | 52.10 / 35.61           | 53.15 / 12.38              | 63.72 / 14.56               | 79.21 / 55.67      | 64.25 / 42.87      |
> | ConVIRT | MIMIC-CXR (filtered)     | 53.85 / 36.45           | 53.80 / 13.25              | 63.51 / 14.73               | 80.15 / 56.10      | 65.70 / 44.10      |
> | ConVIRT | SynCXR                  | 59.49 / 40.51           | 56.07 / 15.43              | 63.43 / 15.10               | 82.08 / 58.38      | 75.55 / 57.43      |

---

> > ### Comment · Reviewer_34ZS · 2024-11-25
> >
> > Thank the authors for the detailed classification. It addressed some of my concerns. However, I still have some concerns. The scope of this study should be confined to assessing whether a synthetic CXR dataset can improve VLM training. At present, the authors have not incorporated experiments with other modalities, which restricts the broader relevance of the findings. So I decide to keep the original rating.

---

> > > ### Author Response · Authors · 2024-11-25
> > >
> > > Thank you for your response. We are glad to address your concerns.
> > >
> > > - **Focus of Our Work**
> > > As discussed in our response to W5, our work focuses on demonstrating the effectiveness of using purely synthetic data to train a robust MedVLP model. Our aim is not to design a new image/text generative model. Therefore, we use only off-the-shelf, publicly accessible models to construct our generation pipeline.
> > >
> > > - **Rationale for CXR Modality**
> > > Currently, the chest X-ray (CXR) modality is the only one with a generative model verified by clinicians, making it usable in practice. Once generative models are proposed and validated for other modalities, our approach can be directly adapted to these new modalities.
> > >
> > > - **Request for Consideration**
> > > We sincerely appreciate your feedback. If possible, we kindly request the reviewer to consider increasing the score to a positive rating, such as 6. This would encourage further exploration of novel generative model designs and foster advancements in synthetic data learning within the community.

---

> > > ### Author Response · Authors · 2024-11-27
> > >
> > > Thank you again for your thoughtful feedback. We kindly invite you to review our new rebuttal, where we have thoroughly addressed your concerns. We hope it provides clarity and resolves the issues raised, and we remain open to further discussion or additional suggestions to ensure all points are adequately addressed.
> > > We appreciate your time and consideration.

---

### Official Review · Reviewer_swT7 · 2024-10-31

**Soundness:** 3
**Presentation:** 1
**Contribution:** 3
**Rating:** 5
**Confidence:** 5

**Summary:**

This paper details a pipeline for generating fully synthetic datasets with the downstream objective of pretraining large vision-language models (VLMs). The synthetic datasets are generated semi-automatically by producing a set of entities using RaTE, a named entity recognition algorithm, which are sampled to produce a subset of findings per sample and are then used to generate a textual report. The textual report is iterated with LLMs to ensure congruence with the initial subset of entities, and is then used for the generation of synthetic images using a text-to-image algorithm. A series of datasets are created using this methodology, which are then explored in a battery of different evaluation strategies for zero-shot classification & segmentation via grounding and further fine-tuning with real data to improve initial VLM performance. The extensive numerical results show consistent performance improvement in VLM when training with mixed real and synthetic data.

**Strengths:**

•	Fully synthetic image generation might open the door to more balanced datasets for training larger vision models, which is a challenging topic in medical image analysis.
•	The evaluation presented is extensive and covers many different aspects of image analysis in the medical domain
•	The work delves into recent trends in AI, where pre-trained services are connected to create outcomes beyond their original intended usage (i.e., using pretrained LLMs, NER algorithms, and TTI synthesis…).
•	The emphasis on only using open-source models is commendable.

**Weaknesses:**

•	The evaluation as expressed in the document is rather confusing, where in many cases it’s not clear which is the evaluation/test set used to derive a set of results (e.g., Table 4, Fig 3).


•	There is a lack of a discussion section, which affects the completeness of the work. It would be preferred to move certain aspects of the methodology (and even some results!) to the appendix (e.g., the specific prompts used to identify and filter low-quality images) and create a proper discussion. The reason behind this is because there are some aspects of this work which puzzle the reader (e.g., the low F1 scores in most zero-shot evaluation techniques) that could/should be addresed in the discussion.


•	Section 3.2. could be worded better, so that it’s clear that a single sample contains both anatomical and clinical entities. I found it slightly confusing


•	There are many aspects of this work that would need remediation or discussion. In no particular order:

1.	In the “analysis” section, there is no description of which test set is employed, so these performance reports are difficult to interpret, validate and compare.

2.	Section 5 should be merged with 4.2. to create a singular “results” section.

3.	Why do you claim “Some methods also leverage external datasets to boost performance, raising concerns about generalizability”? No metric has been provided.

4.	Why do you claim that “Our results confirm that many images in the dataset exhibit these issues” in section 3.1.? No metric has been provided.

5.	Much of the evaluation focuses on classification – what are we to gain using a VLM as opposed to a self-supervised CXR model like Rad-DiNO?

6.	You’ve identified 150k+ entities, but the synth dataset is 200k. This would lead to the generation of a very low number of samples-per-entity. This can also lead to illogical combinations (e.g., pneumothorax in rib) which AFAIK are not accounted for.

7.	The generation efforts described in 3.2. look quite compute-intensive. It’s not essential for the work but I’d be interested to have some approximate time taken to generate a single sample (i.e., text generation + filtering and QA + IMPRESSION generation + TTI).

8.	This methodology could easily lead to collapse because of feeding an algorithm with synth data, see 10.1038/s41586-024-07566-y

9.	The F1 and Dice/IoU scores are quite low in most zero-shot evaluation techniques (tables 1 and 2). It could use some discussion/comparison to other vanilla VLMs and non-VLMs (e.g., zero-shot segmentation with Rad-DiNO). Moreover, the IoU increase wrt the real baseline is not super significant.

10.	It’s not clear how exactly the models are fine-tuned in Section 4.2. What’s the pretraining dataset? Which is the dataset used for fine-tuning?

11.	There’s no comparison to other concept generation or filtering techniques, which is the main contribution of your work. You provide a large amount of experimental runs for downstream applications but the bulk of your work is the synthetic data generation pipeline. Although this is understandable because a paper has to be constrained somehow, some mention in discussion/future steps would be required.

12.	Section 5 contains claims that have not been formally and are hipotheses, e.g.: “likely due to the persistent long-tailed distribution, as the synthetic reports are generated based on real images” or “as synthetic images can be curated using our image filtering procedure to ensure high quality, avoiding issues commonly found in real datasets”.

**Questions:**

See weakness part and please answer the listed question and concerns

---

> ### Author Response · Authors · 2024-11-23
>
> > **W1, W4-1:** Evaluation Details on Table 4 and Figure 3
> - In the experiments presented in Table 4 and Figure 3, the "average zero-shot classification AUC" represents the average performance across all zero-shot classification results, as detailed in Section 4.1. The datasets and implementation details for the zero-shot evaluation are provided in Appendix A.1 and A.2.
>
> > **W2, W3, W4-2:** Content Modification and Expression in Section 3.2
> - Thank you for your feedback. We will revise the content and improve the expression in Section 3.2 in the camera-ready version after acceptance.
>
> > **W2:** Lower F1 Scores in Zero-Shot Evaluation
> - First, we directly refer to the zero-shot performance of the baseline methods as reported in [1]. Furthermore, we strictly re-pretrained all baseline methods on our SynCXR dataset following their official codebases. This ensures that we did not alter their training frameworks but simply replaced the real data with our synthetic dataset to evaluate the effectiveness of the synthetic data. Therefore, the lower F1 scores are attributable to the limitations of the baseline methods themselves, not to our synthetic dataset.
>
> [1] Phan, Vu Minh Hieu, et al. "Decomposing Disease Descriptions for Enhanced Pathology Detection: A Multi-Aspect Vision-Language Pre-training Framework." *Proceedings of the IEEE/CVF Conference on Computer Vision and Pattern Recognition*, 2024.

---

> ### Author Response · Authors · 2024-11-23
>
> > **W4-3:** Using External Datasets to Boost VLP
> - In our related work, we note that "some methods also leverage external datasets to boost performance, raising concerns about generalizability." We elaborate on this in Section 3.3, where we explain that methods such as MedKLIP, KAD, and MAVL rely on external datasets like RadGraph, which are annotated by humans. Additionally, MAVL requires human re-annotation of reports, making these approaches neither scalable nor generalizable due to the high cost of human annotation. Moreover, there is no metric to define or standardize which external datasets are used.
>
> [1] Wu, Chaoyi, et al. "Medklip: Medical knowledge enhanced language-image pre-training for x-ray diagnosis." Proceedings of the IEEE/CVF International Conference on Computer Vision. 2023.
>
> [2] Zhang, Xiaoman, et al. "Knowledge-enhanced visual-language pre-training on chest radiology images." Nature Communications 14.1 (2023): 4542.
>
> [3] Phan, Vu Minh Hieu, et al. "Decomposing Disease Descriptions for Enhanced Pathology Detection: A Multi-Aspect Vision-Language Pre-training Framework." Proceedings of the IEEE/CVF Conference on Computer Vision and Pattern Recognition. 2024.
>
> > **W4-4:** Metric for Identified Data Issues
> - We successfully identified over 7,000 problematic images, accounting for more than 3% of the real dataset. However, thoroughly detecting all problematic images is impractical as it would require manual verification of each image at significant cost. Our goal is to highlight and address issues in the real dataset, not to create a perfectly clean dataset.
>
> > **W4-5:** Comparison with Rad-DINO
> - Rad-DINO was pre-trained on 880k images with a resolution of 518 × 518 from multiple CXR datasets, as reported on the official model page. In contrast, the baseline methods we implemented strictly use the MIMIC-CXR dataset, consisting of 210k images with a resolution of 256 × 256. This disparity in dataset size (four times larger) and resolution (double) makes a fair comparison between our Vision-Language Model (VLM) and Rad-DINO challenging.
>
> - Additionally, Rad-DINO does not provide fine-tuning code or downstream task data splits, which prevents us from directly comparing our VLM under Rad-DINO’s settings.
>
> - Since Rad-DINO is pre-trained with image-only data, it cannot perform zero-shot classification as it lacks alignment with textual information. To evaluate Rad-DINO, we implemented fine-tuning classification using MGCA’s [1] ViT fine-tuning strategy and applied the same data split as our work.
>
> - The results, shown in the table below, indicate that even though Rad-DINO was pre-trained on a significantly larger dataset than MedVLP, the MedVLP method with our synthetic data still outperforms Rad-DINO. This is because VLP incorporates semantic information from textual reports, enabling richer contextual understanding, whereas Rad-DINO, relying solely on visual SSL, focuses only on visual invariances without capturing clinical relevatn semantic information.
>
> | Dataset (Data Ratio) | RSNA (1%) | RSNA (10%) | RSNA (100%) | SIIM (1%) | SIIM (10%) | SIIM (100%) | Covid19 CXR-2 (1%) | Covid19 CXR-2 (10%) | Covid19 CXR-2 (100%) | ChestXray-14 (1%) | ChestXray-14 (10%) | ChestXray-14 (100%) |
> |-----------------------|-----------|------------|-------------|-----------|------------|-------------|---------------------|---------------------|----------------------|-------------------|--------------------|---------------------|
> | Rad-DINO             | 78.27     | 84.51      | 86.32       | 72.40     | 80.45      | 90.80       | 90.40              | 96.75              | 98.70               | 56.42            | 73.15             | 79.65              |
> | ConVIRT-Syn          | 79.01     | 85.58      | 87.90       | 73.51     | 81.10      | 91.84       | 91.50              | 98.80              | 99.73               | 57.45            | 73.60             | 80.20              |
> | GLORIA-Syn           | 80.30     | 86.75      | 88.00       | 76.01     | 87.40      | 92.11       | 94.01              | 98.41              | 99.75               | 60.11            | 74.01             | 81.11              |
>
>
> [1] Wang, Fuying, et al. "Multi-granularity cross-modal alignment for generalized medical visual representation learning." Advances in Neural Information Processing Systems 35 (2022): 33536-33549.

---

> ### Author Response · Authors · 2024-11-23
>
> > **W4-6:** Potential Issues in Generating Reports from Sampled Entities
> - First, we identified five groups of entities: 55,047 Abnormality, 36,365 Non-Abnormality, 23,017 Disease, 22,103 Non-Disease, and 40,517 Anatomy entities, totaling over 150k. We sample entities independently from each group rather than from the entire pool of 150k entities at once.
>
> - For illogical combinations, such as "pneumothorax in rib," the pre-trained LLM, having been trained on a large corpus that includes medical data, can address these inconsistencies. For example, querying the LLM with "Can pneumothorax occur in the rib?" would result in a negative answer. This demonstrates the LLM's ability to understand relationships between entities and avoid generating unreasonable combinations.
>
> > **W4-7:** Time Consumption in Synthetic Data Generation
> - We use the **vLLM** [1] package to accelerate text generation, filtering, QA, and impression generation stages. For Text-to-Image (TTI) generation, we use the **diffusers** package and the official code provided by **RoentGen** [2]. Below, we list the time required for each step in processing a single sample.
>
> - To benchmark the generation process, we used **Llama3.1-8B-Instruct** for report and impression generation and **InternVL2-26B** for image filtering and QA. Note that the times listed below are for single-sample inference. In practice, **vLLM** optimizes batch processing by selecting the optimal batch size, so the total time for processing the dataset can be less than the product of the number of samples and the per-sample time. All evaluations were performed using a single A100 GPU.
>
> | Procedure Name           | Consuming Time/Each Sample (s) |
> |--------------------------|---------------------------------|
> | Report Generation        | 0.3                            |
> | Image Filtering with QA  | 0.1                            |
> | Impression Generation    | 0.2                            |
> | Text-to-Image Generation | 0.7                            |
>
> [1] https://github.com/vllm-project/vllm
> [2] Bluethgen, Christian, et al. "A vision–language foundation model for the generation of realistic chest x-ray images." *Nature Biomedical Engineering* (2024): 1-13.
>
> > **W4-8:** Potential Collapse Risk with Synthetic Data Training
> - The paper [1], which claims that recursively training AI models with synthetic data can lead to model collapse. This process involves generating synthetic data \( D_{A_0} \) with model \( A_0 \), training \( A_0 \) on \( D_{A_0} \) to produce an updated model \( A_1 \), then using \( A_1 \) to generate new synthetic data \( D_{A_1} \), and repeating the process iteratively.
>
> - However, our pipeline fundamentally differs from this recursive approach. We use publicly available models, specifically **InternVL** and **Llama**, to generate the synthetic data, which is then used to train our MedVLP model. Importantly, we do not use MedVLP—or any model trained on synthetic data—to generate additional synthetic data. Since we do not recursively use synthetic data for both generation and training, the concerns described in [1] do not apply to our methodology.
>
>
> [1] Shumailov, Ilia, et al. "AI models collapse when trained on recursively generated data." *Nature* 631.8022 (2024): 755-759.
>
> > **W4-9:** Explanation for Low F1 and Dice/IoU Scores
> - As noted in our response to **W2**, we directly reference the zero-shot performance of baseline methods as reported in [1]. We also strictly re-pretrained all baseline methods on our SynCXR dataset using their official codebases. This ensures that we did not modify their training frameworks but only replaced the real data with synthetic data to evaluate its effectiveness. Therefore, lower F1, Dice, or IoU scores are due to the limitations of the baseline methods, not our synthetic dataset.
>
> - Furthermore, as shown in Table 2(b), our results demonstrate improvements in IoU scores when using synthetic data: +1.89 for ConVIRT and +4.66 for mixed data, compared to models pre-trained on the real dataset alone. This indicates that MedVLP benefits significantly from our synthetic dataset.
>
> [1] Phan, Vu Minh Hieu, et al. "Decomposing Disease Descriptions for Enhanced Pathology Detection: A Multi-Aspect Vision-Language Pre-training Framework." *Proceedings of the IEEE/CVF Conference on Computer Vision and Pattern Recognition*, 2024.
>
> > **W4-10:** Pretraining and Fine-tuning Datasets in Section 4.2
> - For pretraining, we used both the MIMIC-CXR dataset (referred to as the "real dataset") and the SynCXR dataset (generated by our pipeline). The "mixed dataset" combines MIMIC-CXR and SynCXR, without additional curation.
> - For the fine-tuning dataset, we provide detailed information in Appendix A.1 and A.2.

---

> ### Author Response · Authors · 2024-11-23
>
> > **W4-11:** Comparison to Other Concept Generation or Filtering Techniques
> - First, there are no existing methods that use purely synthetic data to train a MedVLP model. We are the first to explore this approach.
>
> - There is only one method, SynthCLIP [1], that uses purely synthetic image-text pairs to train a CLIP model based on natural image-text pairs. However, their approach has limitations, as it generates text with only one entity per sample. In the medical domain, real-world data typically contains multiple entities in a single sample, as explained in Section 3 of our main article.
>
> - Therefore, a direct comparison is impractical due to the differences in methodology and the domain-specific requirements of medical data.
>
> [1] Hammoud, Hasan Abed Al Kader, et al. "SynthCLIP: Are We Ready for a Fully Synthetic CLIP Training?." *arXiv preprint arXiv:2402.01832* (2024).
>
> > **W4-12:** Clarifying Claims in Section 5
> - For the claim, *"likely due to the persistent long-tailed distribution, as the synthetic reports are generated based on real images"*: This is straightforward. As detailed in the last part of Section 3.1, we successfully detected the long-tailed distribution in the real dataset. Since the reports are paired with the images, the long-tailed distribution in the reports reflects the same distribution in the images. When using a report generation model to generate reports based on these long-tailed images, the resulting reports also inherit this distribution, as they aim to describe the images.
>
> - For the claim, *"as synthetic images can be curated using our image filtering procedure to ensure high quality, avoiding issues commonly found in real datasets"*: This is because synthetic data is generated by models, allowing us to regenerate as many samples as needed. This provides an opportunity to replace low-quality synthetic images with regenerated ones, ensuring the total number of samples remains unchanged. In contrast, for real datasets, removing low-quality images directly reduces the number of available samples, which cannot be replaced, limiting the dataset's size and balance.

---

> ### Author Response · Authors · 2024-11-27
>
> Thank you for all your valuable review comments. We kindly ask you to review our rebuttal, where we have addressed your concerns in detail. We believe it clarifies and strengthens the points raised, and we would be happy to further discuss or address any remaining issues you may have.
>
> Thank you again for your time and consideration.

---

### Official Review · Reviewer_ztHX · 2024-11-02

**Soundness:** 2
**Presentation:** 3
**Contribution:** 3
**Rating:** 8
**Confidence:** 4

**Summary:**

This paper provides a thorough evaluation of the medical visual language model with synthetic data. It is proven that training MedVLM with synthetic data and a carefully designed data generation pipeline can significantly boost the model's performance in multiple downstream tasks. The proposed chest X-ray generation pipeline can generate high-fidelity image-report pairs with the given entity. With proper design, the synthetic data can avoid the data quality issues within the real-world data such as long-tailed distribution, and incorrect or low-quality images.

**Strengths:**

1. The proposed image-report generation pipeline uses entities extracted from the real-world report. By sampling these entities with a balanced prior, the synthetic dataset ensures a more balanced distribution compared with real-world data. This has proven to be helpful in the downstream evaluations.
2. This work has conducted a very thorough evaluation with carefully designed experiment settings. It has evaluated the pre-trained model with different training data on multiple downstream, OOD datasets under different settings. It has also evaluated the influence of using different report generation models and text-guided image generation models, which further validate the design of the proposed pipeline.
3. The paper is well-written and easy to follow, and all the technical details are very clear.

**Weaknesses:**

1. One of the major concerns of this paper is the entity sampling process during report generation. According to the paper, the only constraint on this process is the balanced distribution constraint. However, it is possible that the method samples contradictive entities. For example, “decreased in size” and “increased in size” are two of the top 50 entities in the abnormality entities according to Figure 6. They naturally contradict, but it is possible that the model samples both entities during report generation. Moreover, The quality of the extracted entity is also not very satisfying. There are multiple repeated, or less meaningful entities in the top 200 examples according to the appendix. What makes it worse is that the only quality control on report generation is the evaluation of whether the sampled entity exists or not, rather than if they are reasonable as set. Also, enforcing all the sampled entities to be present in the “Impression” section does not make much sense as well. The “Impression” section in the real-world report may not always contain all the findings. So the quality of the generated report is very concerning without further evaluation, especially considering that there are no example reports provided here. Though it indeed improved the performance.
2. Another secondary concern is about the image quality generated from the synthetic reports. Though the author used a SoTA CXR generative model validated by “clinical experts”, the concerns about generated image correctness still remain, especially considering there might be self-contradicting/incorrect synthetic reports. As a universal concern about the medical image generation method, the method adopted here has no control over the anatomy structure, which means it could generate unrealistic CXR and therefore pollute the downstream training process.
3. Additionally, there are some implementation details missing in the paper. a) The actual value of $k$, $m$, and the sampling threshold $\tau_{max}$ were not mentioned in the paper or appendix, which is very critical to understanding the method. b) There is no specific description of the pre-training setting with mixed data, where the dataset size is doubled, It is unknown if the training time is also doubled or not.
4. Considering that the paper itself focuses on evaluating the influence of synthetic data on MedVLM, 2 baseline methods are not very convincing. Except for the methods mentioned in the paper that are not suitable, the reviewer would still recommend evaluating more baselines like naive CLIP and MGCA[a]. It is necessary to conduct further evaluation to validate the conclusion of this paper.
5. There are some small typos in the paper such as a) Some of the results in Table 3 row 3 (ConVIRT-MIX) are bolded incorrectly. b) The “Medical Image Fine-tuned Classification” section in section A.2 was aligned incorrectly as well. c) Also, it would be better if the author could annotate the exact numbers in Figure 3. The current version is not very clear, especially in the bottom right figure.


### Reference
 - [a] Wang, Fuying, et al. "Multi-granularity cross-modal alignment for generalized medical visual representation learning." Advances in Neural Information Processing Systems 35 (2022): 33536-33549.

**Questions:**

While the improvement shown in the paper is quite impressive and the reviewer actually agreed with the authors about the major conclusion, there are still some major concerns about the proposed data synthetic pipeline, as mentioned above. Also, the reviewer has the following questions:

1. The reviewer wonders if it is possible to conduct any form of data quality evaluation on the synthetic data. Even if the experiment results prove that using the synthetic data has greatly improved the performance, it remains unknown if the incorrect synthetic data has injected incorrect knowledge into the model, considering the loose data generation process.
2. Can the author provide some examples of the generated data in the appendix or supplement? The current paper has no example of the generated data, which makes it less convincing.
3. The reviewer also wonders how well the models may perform if they were trained with the **cleaned** MIMIC-CXR as mentioned in the paper. It seems that this cleaned version of MIMIC-CXR is only used for filtering out low-quality synthetic images.

The reviewer sincerely looks forward to hearing from the author during the rebuttal and would be happy to discuss the concerns mentioned above.

---

> ### Author Response · Authors · 2024-11-23
>
> Thank you for your valuable time and comments. The main concerns are addressed below.
>
> >**W1:** Concerns about entity sampling and constrain on report generation
> - **Potential Contradictory Entities:**
>   Leveraging the rich knowledge embedded in LLMs, these models inherently understand contradictory situations. For example, if we query an LLM with *"Can one anatomical region of the lung have increased size and decreased size at the same region at the same time?"*, the LLM is capable of reasoning that this is impossible because these are opposing processes. The LLM effectively distinguishes between conflicting terms like *"increased size"* and *"decreased size"* by associating them with different entities. This ensures that the generated synthetic report remains logical and avoids contradictory entities.
>
> - **Repeated Entities:**
>   We observed repeated entities in the extracted entity set. To address this, we utilized Med-CPT [1], a text embedding model, to compute text embeddings for all entities. Next, we calculated the embedding similarity for entities within each group: `ABNORMALITY`, `NON-ABNORMALITY`, `DISEASE`, `NON-DISEASE`, and `ANATOMY`. For entities with an embedding similarity greater than 0.95, we identified one entity as redundant and replaced it with the other. After this process, each group was refined, resulting in a total of **98,758 unique entities**. Approximately **36% of entities** were removed from the initial pool. We then re-implemented our synthetic data generation procedure based on this de-duplicated set of entities.   \
> \
> Subsequently, we re-pretrained **ConVIRT** and **GLoRIA** on the synthetic dataset generated from the de-duplicated entities. As shown in the zero-shot classification results below, the synthetic dataset consistently improves both MedVLP methods on downstream tasks. We appreciate the reviewers' valuable suggestions, which have inspired us to further enhance the quality of our synthetic data generation pipeline.
>
> | Method  | Pre-training Data | CheXpert (AUC ↑ / F1 ↑) | ChestXray-14 (AUC ↑ / F1 ↑) | PadChest-seen (AUC ↑ / F1 ↑) | RSNA (AUC ↑ / F1 ↑) | SIIM (AUC ↑ / F1 ↑) |
> |---------|-------------------|--------------------------|-----------------------------|------------------------------|---------------------|---------------------|
> | ConVIRT | MIMIC-CXR         | 52.10 / 35.61           | 53.15 / 12.38              | 63.72 / 14.56               | 79.21 / 55.67      | 64.25 / 42.87      |
> | ConVIRT | SynCXR            | 59.49 / 40.51           | 56.07 / 15.43              | 63.43 / 15.10               | 82.08 / 58.38      | 75.55 / 57.43      |
> | ConVIRT | SynCXR(de-duplicated)     | 60.12 / 41.03           | 56.75 / 15.88              | 64.15 / 15.43               | 82.95 / 59.20      | 76.33 / 57.90      |
> | GLoRIA  | MIMIC-CXR         | 54.84 / 37.86           | 55.92 / 14.20              | 64.09 / 14.83               | 70.37 / 48.19      | 54.71 / 40.39      |
> | GLoRIA  | SynCXR            | 61.38 / 41.05           | 57.47 / 15.60              | 64.26 / 15.02               | 72.34 / 49.50      | 67.32 / 53.86      |
> | GLoRIA  | SynCXR(de-duplicated)     | 62.01 / 41.58           | 58.03 / 16.12              | 64.93 / 15.45               | 73.12 / 50.12      | 68.05 / 54.33      |
>
>
> - **Enforcing the Generation of the Impression Section:**  We focus on generating the "Impression" section of reports because the generative model, RoentGen [1], is trained exclusively on text conditioned by "Impression" data. As a result, this model is designed to accept only "Impression"-style text as input. Additionally, in the MIMIC-CXR dataset, which serves as the reference real-world dataset in this work, there are 213,384 image-text pairs that include an "Impression" section, compared to only 137,485 samples that include a "Findings" section. This discrepancy indicates that the "Impression" section is more commonly present in radiology reports than the "Findings" section, making it the logical focus for our generation pipeline.
>
>
> [1] Jin, Qiao, et al. "MedCPT: Contrastive Pre-trained Transformers with large-scale PubMed search logs for zero-shot biomedical information retrieval." Bioinformatics 39.11 (2023): btad651.
> [2] Bluethgen, Christian, et al. "A vision–language foundation model for the generation of realistic chest x-ray images." Nature Biomedical Engineering (2024): 1-13.

---

> ### Author Response · Authors · 2024-11-23
>
> >**W2:** Concerns About the Quality of Synthetic Images
>
> - **Imperfect Synthetic Images:**
>   We acknowledge that synthetic images may contain inaccuracies. However, these images are used solely during the vision-language pretraining stage, where the presence of noisy data is not uncommon. For instance, the CLIP-LAION2B dataset includes noisy samples, and the MIMIC-CXR dataset, as demonstrated in our work and noted in BioViL-T [1], contains many unpaired image-text samples. Despite these imperfections, VLP methods pre-trained on both datasets have demonstrated strong performance on downstream tasks involving natural and medical images [2,3]. This suggests that Vision-Language Pretraining models can effectively learn representative features even when the data contains noise or when images and texts are not perfectly aligned. Therefore, while some synthetic images in our dataset may be imperfect or even contradict their associated reports, they are unlikely to significantly affect the VLP process.
>
> - **Anatomical Region Control:**
>   We agree that fine-grained regional control, such as employing models like ControlNet, could improve the correctness of synthetic images by ensuring better alignment with anatomical structures. However, as of this project's completion, implementing such control faces two key challenges: (1) the absence of fine-grained segmentation tools capable of generating accurate masks for individual anatomical regions, and (2) the lack of paired chest X-ray mask-text datasets required to train ControlNet or similar models for fine-grained regional control. Addressing these gaps remains a critical area for future research to advance synthetic medical image generation.
>
> [1] Bannur, Shruthi, et al. "Learning to exploit temporal structure for biomedical vision-language processing." Proceedings of the IEEE/CVF Conference on Computer Vision and Pattern Recognition. 2023.
>
> [2] Wang, Fuying, et al. "Multi-granularity cross-modal alignment for generalized medical visual representation learning." Advances in Neural Information Processing Systems 35 (2022): 33536-33549.
>
> [3] Radford, Alec, et al. "Learning transferable visual models from natural language supervision." International conference on machine learning. PMLR, 2021.

---

> ### Author Response · Authors · 2024-11-23
>
> > **W3:** Implementation Details of Generation and VLP
> - We set *k=9*, *m=3*, and *$τ_{max}=15$* in our work.
>
> - **Specific Description of tVLP with Mixed Data:**
>   We implemented the baseline methods on mixed data following their official codebases [1][2]. The only modification was increasing the dataset size, which resulted in doubling the training time.
>
> [1] Zhang, Yuhao, et al. "Contrastive learning of medical visual representations from paired images and text." Machine Learning for Healthcare Conference. PMLR, 2022.
>
> [2] Huang, Shih-Cheng, et al. "Gloria: A multimodal global-local representation learning framework for label-efficient medical image recognition." Proceedings of the IEEE/CVF International Conference on Computer Vision. 2021.

---

> ### Author Response · Authors · 2024-11-23
>
> > **W4:** Evaluation on MGCA and Naive CLIP
> - Thank you for your suggestion. We re-implemented MGCA [1] and Naive CLIP [2] using their official codebases on both our SynCXR dataset and its de-duplicated version. The results are presented below. As shown in the table, both VLP methods pre-trained on our SynCXR dataset outperform those pre-trained on the real dataset. This highlights the effectiveness of our data generation framework.
>
>
> | Method  | Pre-training Data         | CheXpert (AUC ↑ / F1 ↑) | ChestXray-14 (AUC ↑ / F1 ↑) | PadChest-seen (AUC ↑ / F1 ↑) | RSNA (AUC ↑ / F1 ↑) | SIIM (AUC ↑ / F1 ↑) |
> |---------|---------------------------|--------------------------|-----------------------------|------------------------------|---------------------|---------------------|
> | CLIP    | MIMIC-CXR                 | 50.17 / 34.11           | 52.15 / 12.25              | 65.52 / 15.69               | 77.58 / 54.12      | 62.78 / 41.82      |
> | CLIP    | SynCXR                    | 58.67 / 38.87           | 54.51 / 15.26              | 64.76 / 15.17               | 82.08 / 60.36      | 77.02 / 57.96      |
> | CLIP    | SynCXR(de-duplicated)     | 59.69 / 39.43           | 55.69 / 16.52              | 65.73 / 16.59               | 82.74 / 60.98      | 77.84 / 58.73      |
> | MGCA    | MIMIC-CXR                 | 53.93 / 38.08           | 56.70 / 15.21              | 64.95 / 14.59               | 71.01 / 48.82      | 55.47 / 39.49      |
> | MGCA    | SynCXR                    | 59.83 / 42.77           | 57.43 / 17.59              | 64.53 / 16.93               | 73.75 / 49.16      | 67.87 / 53.65      |
> | MGCA    | SynCXR(de-duplicated)     | 60.94 / 43.62           | 57.58 / 18.55              | 65.69 / 17.54               | 74.92 / 50.47      | 68.91 / 54.79      |
>
>
> > **W5:** Typos Correction
> - Thank you for your detailed revision. We have corrected the typos and updated the figures, and these changes will be reflected in the camera-ready version.
>
> [1] Wang, Fuying, et al. "Multi-granularity cross-modal alignment for generalized medical visual representation learning." Advances in Neural Information Processing Systems 35 (2022): 33536-33549.
>
> [2] Radford, Alec, et al. "Learning transferable visual models from natural language supervision." International conference on machine learning. PMLR, 2021.
>
> > **Q1:** Data Quality Check on Synthetic Data
> - To assess the quality of the synthetic data, we used BiomedCLIP [1], a foundational vision-language model, to compute the CLIP score for all samples in the synthetic dataset and then averaged the results. The CLIP score measures the alignment between synthetic reports and images. Similarly, we computed the averaged CLIP score for the real dataset as a reference.
> \
>   The results show that the averaged CLIP score for the real dataset is *0.74*, while our SynCXR dataset achieves a score of *0.72*. This close alignment demonstrates that our synthetic dataset exhibits a level of coherence and quality comparable to the real dataset, underscoring the robustness of our data generation framework.
>
>
> [1] Zhang, Sheng, et al. "BiomedCLIP: a multimodal biomedical foundation model pretrained from fifteen million scientific image-text pairs." arXiv preprint arXiv:2303.00915 (2023).
>
> > **Q2:** Synthetic Samples
> - Thank you for your suggestion. Yes, we will update the synthetic reports and images in the camera-ready version.

---

> ### Author Response · Authors · 2024-11-23
>
> > **Q3:** Results on Cleaned MIMIC-CXR
> - We re-pretrained ConVIRT on the cleaned version of MIMIC-CXR, and the results are shown below. As the table demonstrates, the VLP method pre-trained on the cleaned MIMIC-CXR dataset slightly outperforms the version pre-trained on the uncleaned dataset. However, it still underperforms compared to the method pre-trained on our synthetic data. This can be attributed to two main reasons:
>
>   1. Even after filtering out many imperfect samples from the real dataset, it is impractical to thoroughly remove all poor or unpaired samples. As a result, some problematic samples remain in the real dataset, negatively impacting the VLP process.
>
>   2. The cleaning process for MIMIC-CXR primarily focused on filtering samples based on image quality. However, the long-tailed distribution of entities in the dataset was not addressed, as simply downsampling or oversampling image-text pairs is not feasible. This issue limits the representational balance in the cleaned dataset.
>
>
>
> | Method  | Pre-training Data       | CheXpert (AUC ↑ / F1 ↑) | ChestXray-14 (AUC ↑ / F1 ↑) | PadChest-seen (AUC ↑ / F1 ↑) | RSNA (AUC ↑ / F1 ↑) | SIIM (AUC ↑ / F1 ↑) |
> |---------|-------------------------|--------------------------|-----------------------------|------------------------------|---------------------|---------------------|
> | ConVIRT | MIMIC-CXR               | 52.10 / 35.61           | 53.15 / 12.38              | 63.72 / 14.56               | 79.21 / 55.67      | 64.25 / 42.87      |
> | ConVIRT | MIMIC-CXR (cleaned)     | 53.85 / 36.45           | 53.80 / 13.25              | 63.51 / 14.73               | 80.15 / 56.10      | 65.70 / 44.10      |
> | ConVIRT | SynCXR                  | 59.49 / 40.51           | 56.07 / 15.43              | 63.43 / 15.10               | 82.08 / 58.38      | 75.55 / 57.43      |
> | ConVIRT | SynCXR (de-duplicated)  | 60.12 / 41.03           | 56.75 / 15.88              | 64.15 / 15.43               | 82.95 / 59.20      | 76.33 / 57.90      |

---

> ### Comment · Reviewer_ztHX · 2024-11-23
> **Reply to Author's Rebuttal**
>
> I sincerely appreciate the author's feedback and it is very exciting to see all those extra experimental results with promising outcomes. Overall, the author's feedback addressed most of my concerns and questions.
>
> Still, there are 3 minor questions.
> - First of all, the fact that LLM can understand the relationship between each contradictive entity does not necessarily mean it can avoid this issue when contradictive entities are provided in the input. LLM can still generate problematic captions simply using all the given entities rather than figure out if they are reasonable or not. The reviewer believes it is still necessary to double-check this problem, and maybe take a closer look into the generated captions.
> - Meanwhile, the new results after filtering out the repeated entities improved the performance of the pipeline and this may indicate a major revision in terms of the paper's content is needed to reflect this and further clarify the whole image synthesis pipeline.
> - Lastly, I largely agree with the author's comments on the synthesis image quality issue. However, it concerns me without qualitative or quantitative evidence.
>
> These issues put me in a hard position, I would like to recommend acceptance for the paper given its contribution and the inspiring results, but it seems that there are still things that can be further optimized in the image synthesis pipeline, which makes it slightly below the bar of 8: Directly accept. I would really like to raise my score to 7: weakly accept, but there is no such a number here. So I choose to maintain my current score of 6 for now and wait to see what other reviewers think.

---

> > ### Author Response · Authors · 2024-11-24
> >
> > We deeply appreciate the positive feedback from the reviewer and are happy to discuss the further minor questions:
> >
> > - **Hallucination in LLM-Generated Captions:**  We acknowledge the importance of addressing imperfections in captions generated by LLMs, as hallucination remains an open challenge in the ML community. However, we note that this issue can be mitigated with the development of more powerful LLMs [1]. This indicates that improvements in caption quality can benefit from advancements in the broader LLM community as new, more capable models are developed.
> >
> >    [1] Li, Junyi, et al. "HaluEval: A Large-Scale Hallucination Evaluation Benchmark for Large Language Models." *Proceedings of the 2023 Conference on Empirical Methods in Natural Language Processing*, 2023.
> >
> > - **De-duplicated Entity Procedure:**  We are sincerely grateful to the reviewer for highlighting this point, which has helped us refine our work. We will add a discussion in Section 3.1 (Entity Mining) of the main article to emphasize the importance of the de-duplicated entity procedure. However, we wish to clarify that this change affects only the input text prompts for image generation and does not alter the image generation pipeline itself.
> >
> > - **Synthetic Data Quality and Future Directions:**  We appreciate the reviewer’s agreement on this point and share the hope that the community will develop large-scale, clinically relevant datasets for training medical ControlNet models to enable fine-grained control in generation. We believe that synthetic data quality can be significantly enhanced as the community progresses in this direction.
> >
> > Finally, we want to express our gratitude once again for the reviewer’s valuable suggestions and active discussion. We sincerely believe that raising your score to an 8 would signal strong support for this work and inspire further exploration and innovation in this promising area. It would also encourage the community to study the use of synthetic data for medical applications, fostering impactful advancements.
> >
> > Thank you again for your thoughtful review and consideration.

---

> > > ### Comment · Reviewer_ztHX · 2024-11-26
> > >
> > > Thanks to the author for replying to me again, after carefully reading the review and rebuttal from other reviewers, I decided to raise my final score to 8. I still believe this work demonstrates the promising path toward applying biomedical image synthesis algorithms to medical visual language model training. As suggested in the title, the author has provided an extensive amount of evidence in the specific visual language pre-training field, and the new results during the rebuttal phase also demonstrate this.
> > >
> > > However, I also want to highlight that the pipeline proposed in this work still needs to be further optimized, especially concerning the quality of the generated CXR captions and the correctness of the generated images. This will weaken its soundness as this correctness is desired in real-world applications. Though this may not harm to contribution of this work as the pioneer in this area, it is still a very significant problem. Although my final score is 8, I still want to highlight this concern to ACs and the authors. The reviewer believes this will be the key to better solutions in applying biomedical image synthesis algorithms in real-world scenarios.

---

> > > > ### Author Response · Authors · 2024-11-27
> > > >
> > > > Thank you for increasing the score to 8. This is a tremendous encouragement to us and reinforces our motivation to continue advancing this area of research.
> > > > We agree that a new evaluation system for the quality of synthesized images and text is an important aspect. Since we have demonstrated the effectiveness of synthesized data, we will prioritize this in our future work.
> > > > We deeply appreciate your recognition of our work as a pioneering effort in this field and your constructive feedback, which will guide us toward more robust and practical solutions for real-world applications.

---

### Note · Authors · 2025-01-29

I have read and agree with the venue's withdrawal policy on behalf of myself and my co-authors.